# Procedures and Legal Instruments for Drought Declaration in the Segura River Basin (Spain)

**José Alberto Redondo-Orts \*, Miguel A. Sáez-García \* and María Inmaculada López-Ortiz \***

University Institute of Water and Environmental Sciences, University of Alicante, Carretera San Vicente del Raspeig s/n, 03690 San Vicente del Raspeig, Alicante, Spain
\* Correspondence: jaro@alu.ua.es (J.A.R.-O.); masaez@ua.es (M.A.S.-G.); iortiz@ua.es (M.I.L.-O.)

**Abstract:** The phenomenon of drought and its socioeconomic and environmental consequences have been addressed in many studies, which show that anticipating its diagnosis and activating specific management measures are fundamental for providing an efficient response. In the Segura River Basin, located in south-east Spain, many episodes have occurred throughout history, with devastating effects on production and supply systems. However, they have enabled us to learn and evolve towards developing a resilient system to address these situations, through the application of external resources, transfers from other basins and non-conventional resources derived from the reuse of treated water and desalinated seawater. This evolution has been possible thanks to the advances made in hydrological planning and, specifically, the Special Drought Plans, through the development of indicator systems associated with scenarios which enable the automatic activation of specific actions to reduce the impacts. Climate change is already a reality and has led to an increase in the frequency and intensity of droughts, testing the capacity to respond based on the current policies. Therefore, the objective of this research is to analyse the last drought occurring in the Segura River Basin in the period 2015–2019 by comparing the status indicators developed for detecting drought in the SDP 2007 with its subsequent review carried out in the year 2018, in which these indicators were updated and expanded so as to cover both drought and scarcity. Subsequently, an in-depth analysis has been made of the approved legislation and the measures adopted which consisted in the mobilisation of more than 600 hm³ of extraordinary resources, which have been able to maintain the supply to the population and minimise the economic losses of the productive systems, particularly in irrigated agriculture.

**Keywords:** drought; scarcity; water resources; hydrological planning; water law; socioeconomic impacts; mitigation strategies; agriculture

## 1. Introduction

Water is an essential resource for the socioeconomic and environmental development of a region [1]. Droughts constitute one of the most important and less understood natural risks and are prone to causing significant adverse impacts on this development and on society as a whole [2–8].

Analysing the many previous studies on the definition of drought, we can observe a wide range of perspectives when addressing the issue and discussing its specific description. This has generated one of the principal difficulties for this type of research [9]. However, taking into account the majority of the studies, droughts can be classified into four categories: meteorological drought (significant reduction in precipitations), hydrological drought (insufficient natural resources for the established uses), agricultural drought (soil moisture deficit) and socioeconomic drought (impossibility of satisfying water demands) [10–12].

As a result, drought could be defined as a cyclical negative rainfall variation with an undetermined duration but sufficient to cause a decrease in the available resources to cover human activities and the environment [13,14]. This phenomenon can vary greatly between regions and between countries.

Consequently, drought itself should not be considered as a disaster, as the degree of incidence will depend on the impacts that it generates [15]. These effects are more intense in those regions where there are already imbalances between resources and demands [16].

Traditionally, drought management has been based on identifying the phenomenon as a crisis, directing hydraulic polices towards the construction of large infrastructures aimed principally at satisfying demand and addressing the consequences separately from the causes [17]. It has been shown that the impacts have not been reduced and they have often increased, therefore aggravating the vulnerability of the water systems. For this reason, only the application of policies aimed at reducing risk can increase the resilience to future drought episodes [18]. In this respect, it is necessary to change the response and emergency measures based on the construction of infrastructures and economic compensation, and implement proactive and prevention measures [19] through hydrological planning and collaboration between the different sectors [20].

It is important to note that the concepts of drought (temporary and natural) and scarcity (permanent and anthropogenic) [21] are sometimes used indistinctly. However, water scarcity can be caused or aggravated by situations of drought and other pressures such as the inefficient use of resources or situations of pollution. Therefore, they are different terms, and it is essential to differentiate them in terms of causes, consequences and spheres of application in order to appropriately identify them and address them [22]. Thus, water scarcity should be considered as a situation in which the water resources are not sufficient to satisfy the water demands [1].

Another aspect which should be taken into account is that throughout the world we can observe the impacts generated by climate change [22] on the available water resources in the most vulnerable regions [23], and that scarcity and drought situations could get worse, increasing the area and population living with this water stress [24].

The phenomena of drought and scarcity constitute one of the most important challenges in international water policy, even more so with the exacerbation occurring due to climate change. In the review conducted by this article, these aspects have been examined and related to the Segura River Basin and may be extrapolated to other international basins.

Water scarcity already affects every continent, as water use has been growing globally at more than twice the rate of population increases in the last century, and an increasing number of regions are reaching the limit at which water services can be sustainably delivered, especially in arid regions and growing urban areas [25]. Climate change is also expected to amplify the already complex relationship between world development and water demand [26].

In recent years, considerable efforts and advances have been made on both a scientific and technical level in the European Union to characterise droughts, assess the risk and develop indicators to enable the identification and activation of measures to mitigate their effects. The reduction in water consumption and the adaptation to climate change have concentrated the efforts of the member states, and drought and scarcity have been integrated into sectoral policies [24].

The publication of the Water Framework Directive (hereafter, WFD) in 2000 was one of the most relevant milestones and led to important changes in water management [21]. However, droughts are only addressed tangentially within the WFD, and the elaboration of drought management plans is not mandatory.

Recently, new campaigns have been launched in the European Union that focus on the increase in water scarcity, not only in arid and semi-arid places, with potentially devastating consequences on a global scale if nothing is done about the impact enough to

reverse the situation and increase the risk of the progress to ensure the availability and sustainable management of water and sanitation (Sustainable Development Goal 6) [27].

Spain is a Mediterranean country of the EU in which there is a high level of variability in the spatial and time distribution of water resources, and where scarcity and droughts affect many river basins [28]. The south-east region, the Segura River Basin (hereafter, SRB), is the most affected and constitutes the area of study of this research.

After a series of severe and recurrent drought episodes occurring over the last few decades had been overcome, which had serious economic (agriculture and electricity production) and environmental (worsening of the state of the water bodies) impacts, a change of focus in crisis and risk management took place [29]. One of the most important milestones was the passing of the Law of the National Hydrological Plan (hereafter, NHP) in 2001, which required the elaboration of the Special Drought Plans (hereafter, SDP).

The principal objective of the SDP is to define the relationship between the drought situation in which a river basin is found, and the application of measures through the integration of a system of indicators that enable the automatic activation of the actions to be implemented [30].

The objective of this study is to analyse the last drought occurring in the Segura River Basin in the period 2015–2019. The research analyses the drought indicators developed in the first SDP and its subsequent review carried out in 2018, which is currently in force and in which these indicators were updated and extended in order to cover both drought and scarcity. To do this, a comparison is made of the two methodologies for calculating the indicators and, subsequently, the measures adopted based on the legal provisions passed and their ultimate effectiveness in the management of the risks produced are analysed.

## 2. Field of Study

Spain is one the countries of the European Union with the highest water stress. Water consumption exceeds 40% of the total available resources in 72% of the country's area compared to 26% of the area in Italy or 1% in Germany [31].

With respect to the geographical area of study, the SRB is located in south-east Spain and has an area of 19,025 km² (only the continental part). The territorial area of the SRB covers the Autonomous Region of Murcia and part of the Region of Andalusia (Almería, Granada and Jaén), Castilla-La Mancha (Albacete) and the Region of Valencia (Alicante) as can be observed in Figure 1.

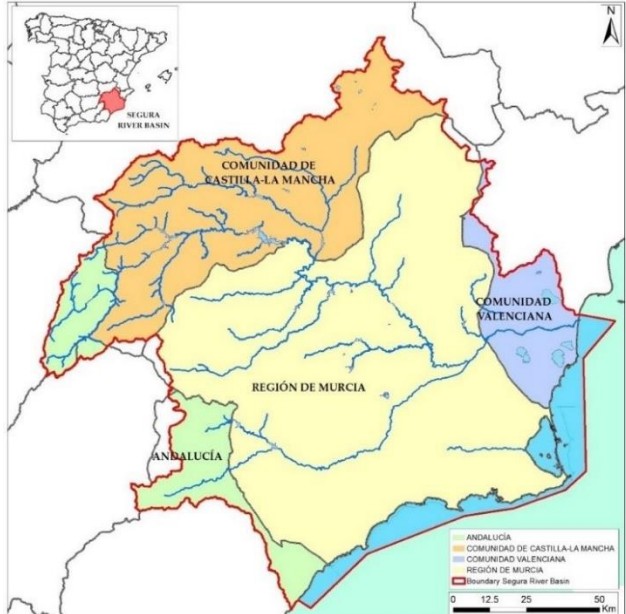

**Figure 1.** Study situation map. Source: Special Drought Plans, own elaboration.

The distribution of rainfall is highly heterogeneous, both spatially and interannually, as the autumn and spring months are characterised by high levels of rainfall while the summer months are dry. By zones, the north-east zone of the basin records more than 1000 mm/year while the coastal zones reach minimum values of less than 100 mm/year. The average estimated rainfall in the Segura Basin Hydrological Plan (hereafter, SBHP) varies between 376 mm/year (series 1940/41–2017/18) and 364 mm/year (series (1980/81–2017/18) [32]. This water stress afflicting the area of study requires a greater efficiency in the use of the available resources.

The complexity of the SRB resides in the comprehensive management of the conventional and non-conventional water resources in a unique operating system [33]. Furthermore, it is the only Spanish basin with structural scarcity [31]. This situation should be studied within the field of hydrological planning, which does not fall within the scope of the objectives of the SDP [34].

The total resources of the SRB have been calculated in the latest plan at 1520 hm³/year (Table 1). The net contributions of the natural system amount to 635 hm³/year (renewable surface water and groundwater, without considering the discharges into the sea), accounting for more than 40% of the total resources. This situation reveals the fragility of the system in drought episodes. The Tajo-Segura Transfer (hereafter, TST) is one of the most important infrastructures of the basin, as the entry of resources from the Tajo basin was designed to partly mitigate the structural deficit of the SRB and ensure a certain security for meeting the demands (urban supply and irrigation). However, after more than 40 years in operation, the average resources transferred only amount to 295 hm³/year [32], of the maximum 600 hm³/year approved in the Law 52/1980 [33], even though these resources have become essential.

Faced with this situation of under-endowment, the strategic importance of the non-conventional resources as a complementary measure should be noted, converting the SRB into one of the most resilient regions, not only during drought episodes but also in situations of normality [35]. Currently, the production capacity of desalination resources is considered to be more than 300 hm³/year (with plans to increase this capacity to 400 hm³/year in the coming years). With respect to reused water, Spain is the country with the highest volume of these resources in the European Union, with 347 hm³/year, accounting for one third of the total of the EU [36]. Within the context of the SRB, both the reused wastewater resources and the returns of irrigation water represented a volume of over 260 hm³/year [32], constituting an example of sustainable management and the circular economy which, in the water sector, consists of using water over and over again, as in the case of the natural cycle [37].

**Table 1.** Total resources of the SRB. Source: own elaboration based on [32].

| Origin of the Resource | Resource (hm³/Year) |
|---|---|
| Natural resources | 635 |
| External transfers [1] | 312 |
| Urban and industrial reuse | 147 |
| Returns of irrigation water | 121 |
| Desalination resources | 305 |
| **TOTAL** | **1520** |

[1] Resources transferred from the Tajo (295 hm³/year) and Negratín (17 hm³/year).

The demands (without considering the environmental demands for the maintenance of humid areas of 32 hm³/year as they are considered a restriction to the system) amount to a value of 1792 hm³/year, distributed between agricultural use 1522 hm³/year (85%), urban use 250 hm³/year (14%), services (irrigation of golf courses) 11 hm³/year (0.5%) and industry unrelated to the supply network of 9 hm³/year (0.5%) [38]. In addition, the following table, (Table 2), shows water consumption (resources that do not return to the

water environment, which evaporate or are incorporated into products) for different uses, which amounts to a total of 1185 hm³/year [39].

The weight of agriculture is highly relevant in the basin in terms of both irrigated crops that represent more than 85% of total demand of the basin, with a net area of more than 260,000 ha (490,000 ha of gross area) and rain-fed crops [40]. The production value associated with irrigation in the Segura Basin exceeds €3000 M/year and the net margin almost €1400 M/year, and the area generates more than 115,000 jobs. On the other hand, the gross value added (hereafter, GVA) in the Segura Basin for the agricultural sector is worth a value of almost €1600 M [41,42].

The agriculture sector of the SRB, specifically irrigated fruit and vegetable crops, is a major exporter and forms the base of a widely developed agro-food sector. The agro-food export figures of the Region of Murcia (principal autonomous region of the SRB) indicate how the exports of 2017 amounted to €4786 M; that is, 11.4% of Spanish agro-food exports and 46% of total exports of the Region of Murcia. Finally, in order to take into account the direct economic importance of the use of water in agriculture in the area of the SRB, it should be noted that the average productivity of irrigated agriculture for 2015 was € 7390/ha, representing 148% of the average value of this indicator calculated for the whole of Spain and the highest of all of the river basins of the Iberian peninsula [43].

**Table 2.** Demands and consumption of water of the SRB. Source: own elaboration based on [32,40].

| Uses | Demands (hm³/Year) | Consumption (hm³/Year) |
|---|---|---|
| Agricultural | 1522 | 1122 |
| Urban | 250 | 52 |
| Services | 11 | 11 |
| Industrial unrelated | 9 | |
| **TOTAL** | **1792** | **1185** |

As described, the demands by far exceed the resources, generating a structural deficit and revealing the sensitivity of the operating system to drought situations [21]. In order to determine this sensitivity, the Water Scarcity and Drought Expert Group of the European Commission presented the WEI (Water Exploitation Index), included within a series of common indicators for water scarcity and drought [44]. A WEI of over 20% indicates the presence of water stress, and over 40% indicates severe scarcity due to strong competition for water and difficulty to maintain the ecosystems [45].

Based on the information of resources, demands and consumptions, two exploitation indices have been calculated that represent the average results of the use of the water consumption in the SRB. The first index (S-WEI) has been obtained by calculating the percentage that demands (1792 hm³/year) represent of the resources (1520 hm³/year), with a value of 118%. The second indicator (WEI+) has been estimated by considering the consumption of water (1185 hm³/year) with respect to the resources (1520 hm³/year), obtaining a value of 85% [46]. As a result, we can conclude that the SRB is in a situation of severe scarcity and the phenomenon is aggravated in times of drought.

The SRB has historically suffered from countless drought periods, which are recorded in the catalogue of historical droughts, elaborated by the Hydrographic Studies Centre of the Centre for Public Works Studies and Experimentation (hereafter, CEH of the CEDEX), for the Directorate General for Water (hereafter, DGW), documenting the droughts occurring prior to 1940 [47]. One of the elements of this report is a database including historical information on 184 drought events, characterised in accordance with their economic, social and hydrological impacts [34].

In order to characterise the droughts after the year 1940, the following figure (Figure 2) represents the contributions regulated in the headwater reservoirs of the SRB from the water year 1940/41 until the last available figure in the inventories of contributions to

headwater reservoirs of the SRB in the water year 2019/20, with an average, for the whole series analysed, of 452 hm³/year [48].

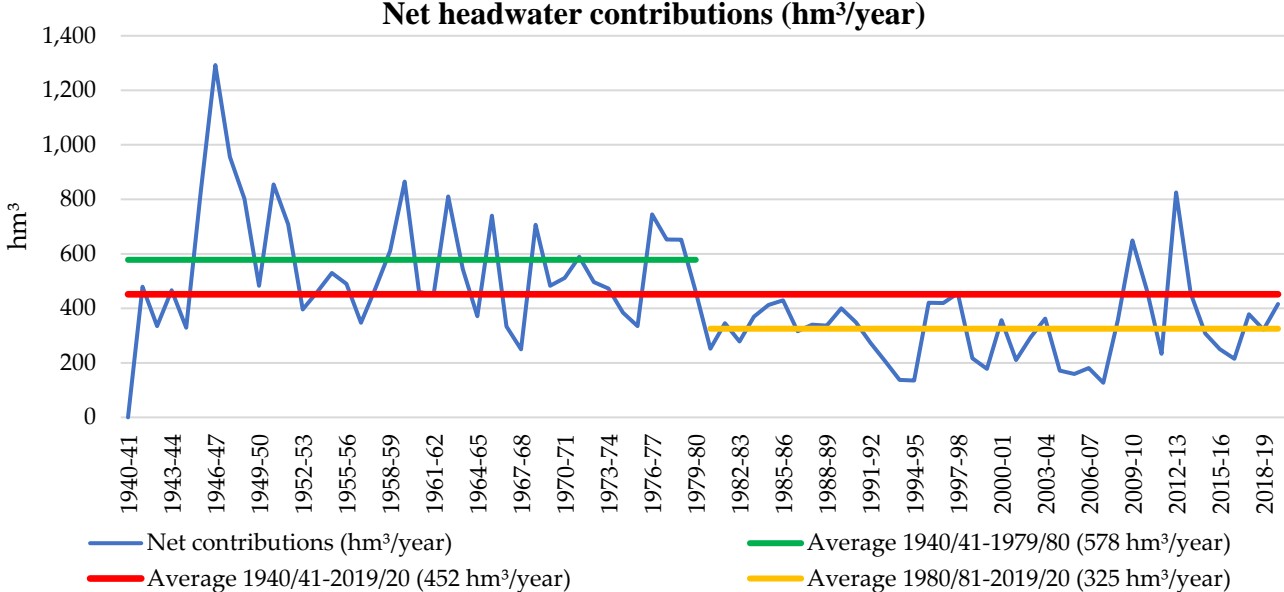

**Figure 2.** Net contributions regulated in the headwater reservoirs of the SRB. Period 1940/41–2019/20. Source: own elaboration based on [49].

In order to carry out a more specific analysis of the variation in contributions, the aforementioned study has been structured into two periods. The first corresponds to the series of the first 40 years from the water year 1940/41 to the water year 1979/80. As we can observe in the following table (Table 3), five episodes of several years have been identified in the period analysed with contributions below the average (578 hm³/year), which can be classified as hydrological droughts.

**Table 3.** Annual net headwater contributions for the droughts detected in the period 1940/41-1980/81. Source: SPD 2018 [34]. Own elaboration.

| Date | Number of Years | Net Headwater Contribution (hm³/Year) |
|---|---|---|
| 1942–1944 | 3 | 335/466/329 |
| 1953–1954 | 1 | 396/460 |
| 1956–1958 | 3 | 490/347/475 |
| 1967–1969 | 2 | 334/250 |
| 1972–1976 | 4 | 496/473/384/335 |

In the second forty-year period analysed, from the water year 1980/81 to the water year 2019/20, the contributions are below the average historical contributions (452 hm³/year), which reflect a significant change in trend and the first symptoms of the change in climate conditions. Although it was a period with lower contributions, the arrival of the water from the Tajo transfer counterbalanced the lower available resources from the Segura in order to meet demand.

In this period, we can identify four drought episodes (1980–1983, 1993–1995, 2005–2008 and 2015–2019), when the contributions were below the average for the period analysed (325 hm³/year). This is reflected in the following table (Table 4).

**Table 4.** Annual net headwater contributions for the droughts detected in the period 1980/81-2019/20. Source: own elaboration based on [34,48].

| Date | Number of Years | Net Headwater Contribution (hm³/Year) |
|---|---|---|
| 1980–1983 | 3 | 252/345/278 |
| 1993–1995 | 3 | 207/138/135 |
| 2005–2008 | 4 | 172/159/181/127 |
| 2015–2019 | 5 | 308/250/215/378/322 |

Figure 3 shows, with greater detail, the final drought period 2015–2019 (indicated in green), including the net headwater contributions, always below the average of the period 1940/41–2019/20 (452 hm³/year) and, except for one year, below that of the period 1980/81–2019/20 (325 hm³/year). As complementary information, the average annual rainfall in the SRB has also been included [49], which fluctuated greatly, together with the resources transferred from the Tajo River which were also at minimum levels [50].

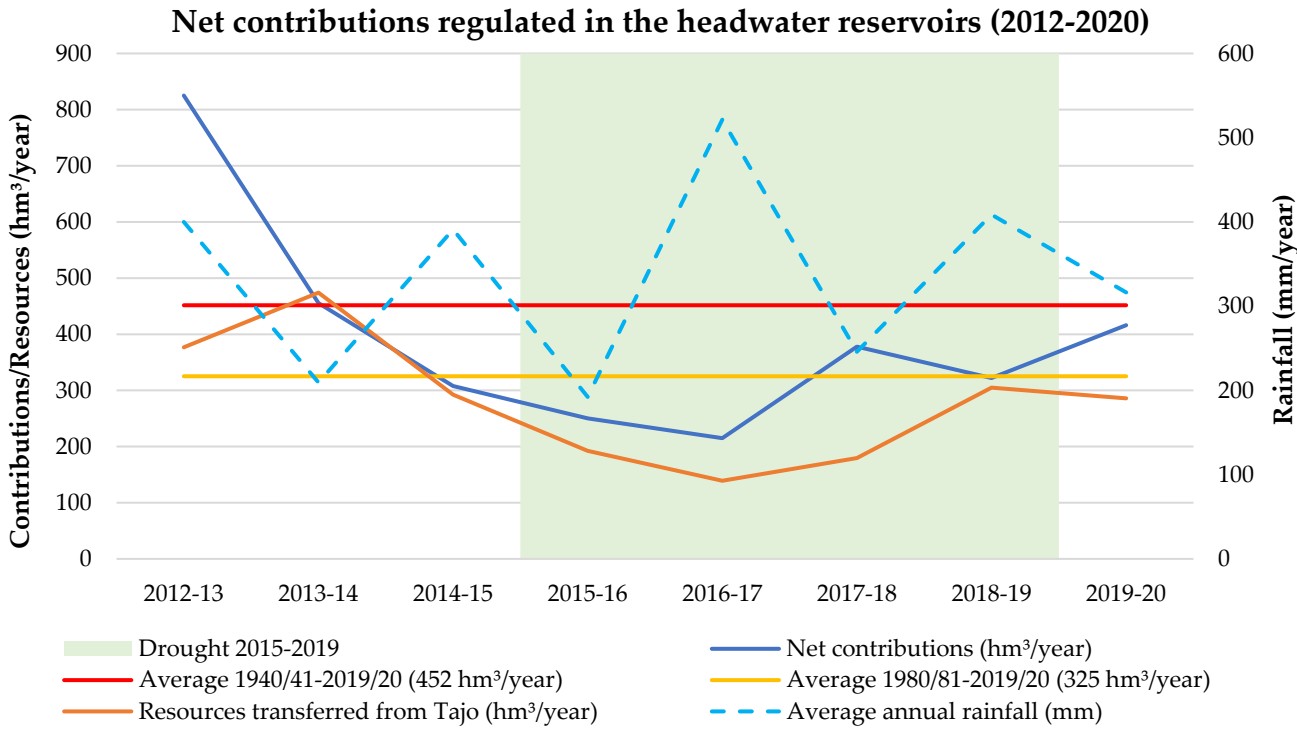

**Figure 3.** Net contributions regulated in the headwater reservoirs of the SRB, average annual rainfall and resources transferred from Tajo, period 2012/13-2019/20. Source: own elaboration based on [48–50].

Based on the drought indicators prevailing in 2015, corresponding to the Special Drought Plan approved in 2007 [51] (hereafter, SDP 2007), a drought was declared in the SRB through the Royal Decree 356/2015 of 8 May, declaring a situation of drought in the territorial area of the Segura Hydrographic Confederation (hereafter, SHC), and exceptional measures were adopted to manage the water resources (in force until 31 December 2015) [52]. Subsequently, four extensions were approved [53–55], with the last in force until 30 September 2019 (Royal Decree 1210/2018, of 28 September) [56].

However, during the period of validity of the Drought Decree, the revision of the Special Drought Plan, in November 2018, was updated and substantially modified and new indicators for quantifying prolonged drought and temporary scarcity were introduced.

Due to the fact that the declaration of the drought in 2015 was made according to the previous SDP 2007, it is necessary to determine whether the indicators defined in the SDP 2018 to identify prolonged drought and temporary scarcity would have been able to adequately detect and anticipate this situation.

The declaration of drought is associated with the activation of measures to mitigate the effects on the demands and ecosystems of the area. A correct and accurate characterisation of these measures is essential.

In view of all of the above, we have carried out an exhaustive analysis of the indicators used in both of the Special Drought Plans and their relationship with the chronology of the drought 2015–2019 in the SRB. We have also contemplated the measures adopted and their repercussions on socioeconomic aspects.

## 3. Materials and Methods

### 3.1. Materials

As established in the National Hydrological Plan (2001) in its Article 27 on drought management, "the organisations of the basin will elaborate within the corresponding Hydrological Plans of the basin special action plans in situations of alert and eventual drought, including the operating rules of the system and the measures to apply in relation to the use of the public hydraulic domain" [57].

In accordance with this mandate, on 21 March 2007, the Special Drought Plan (SDP 2007) in the SRB was approved. Its principal objective was to minimise the environmental, economic and social impacts of drought situations. In order to fulfil this principal objective, a series of specific aims were established, all within the framework of sustainable development [51]:

- To guarantee the availability of water in order to maintain the health and life of the population.
- To prevent and minimise the negative effects of drought on the ecological status of the water bodies, and in particular on the ecological flows, preventing permanent negative effects.
- To minimise the negative effects on the supply of the population and on the economic activities, in accordance with the prioritisation of the Hydrological Plans.

In order to meet these objectives, mechanisms were defined for predicting and detecting drought situations; thresholds of progressive severity phases of droughts were established (normality, pre-alert, alert and emergency), calculated through status indicators; and measures were defined to fulfil the specific objectives in each drought phase, ensuring transparency and public participation at all times [51].

In accordance with the phases established, three levels of measures were contemplated: strategic (pre-alert phase), tactical (alert phase) and emergency (emergency phase). In turn, these actions can be distinguished in terms of their nature as administrative measures, awareness and dissemination measures, supply-related actions (increase in water resources) and demand-related actions (reduction in the demands to satisfy).

On 28 November 2018, the revision of the Special Drought Plan (SDP 2018) was approved, which established a conceptual difference between situations of prolonged drought, associated with the reduction in rainfall and the water resources in a natural regime and the consequences for the natural environment (and therefore, independent from the socioeconomic uses associated to human intervention) and those of temporary scarcity, associated with short-term problems of a lack of resources to meet the demands of the different socioeconomic uses of water. Structural scarcity is not contemplated in the SDP. This occurs when these problems of scarce resources in a specific area are permanent and, therefore, should be analysed and resolved within general hydrological planning [34].

In the SRB, the Standardized Precipitation Index (SPI) has been used to identify prolonged droughts. The SPI is defined as a numerical value that represents the number of

standard deviations of the rainfall throughout the accumulation period of interest with respect to the average. This is the most useful drought indicator as it has the capacity to recognise the importance of the time scales in the analysis of the availability and use of water. Therefore, it can be used in risk assessment and decision making [8,11].

For the case of temporary scarcity, the indicator selected is based on the relationship between the availability of resources and the demands, with the selection of a series of the most representative variables of the evolution of the availability of resources, focusing on the accumulated contributions and the resources stored in reservoirs [34].

Below, we will analyse in detail the two methodologies and their differences in terms of the declaration of drought in the SRB.

First, the methodology developed in the SDP 2008 is examined, which determines the scarcity index used to declare a drought. This methodology is determined both for the basin system (taking into account the contributions and stocks of the headwater reservoirs of the Segura Basin) and the transfer system (taking into account the contributions and stocks of the headwater reservoirs of the Tajo Basin), which, when combined, provide the global status of the basin.

Second, the updated methodology in the SDP 2018 is examined, which determines both the Scarcity Index, with slight modifications with respect to the SDP 2007 and the Prolonged Drought Index, incorporated as a novelty, which uses the 9-month SPI. Finally, combining the two, the conditions for declaring an extraordinary drought are established. This is a new condition in the revised PES 2018.

### 3.2. Methodology SDP 2007

For predicting and detecting drought situations in the SDP 2007, thresholds of progressive severity phases were established based on the calculation of three status indicators: one for the system for exploiting the basin, another for the system for exploiting the transfer and a global indicator for the whole area.

The value of the basin indicators gives greater weight to the headwater contributions of the Segura than the resources stored in the Segura reservoirs, as the drought in the SRB depends more on the contributions than its stored resources. This is related to the high water consumption throughout the year, so the volume of regulation does not allow for much management capacity. The value responds to the following expression:

$$\text{Basin Indicador System} = \frac{2 \cdot \text{Contributions} + \text{Stored resources}}{3} \tag{1}$$

where the contributions are those accumulated over the previous 12 months and the stored resources are those in the principal reservoirs of the Segura Basin (Fuensanta, Cenajo, Camarillas, Talave and Alfonso XIII) on the date of the calculation.

According to the regulations of the Tajo-Segura Transfer [58], the volumes transferable to the Segura depend on the contributions in the Entrepeñas and Buendía reservoirs and the availability of transferable stocks. Therefore, in order to define thresholds and drought status in the transfer system, the following indicator has been considered:

$$\text{Transfer Indicator System} = \frac{\text{Contributions} + 2 \cdot \text{Stored resources}}{3} \tag{2}$$

where the contributions are those accumulated over the previous 12 months in the headwaters of the Tajo and the stored resources are those in the Entrepeñas and Buendía reservoirs (Tajo Basin) on the date of calculation.

The system for exploiting the basin is unique; therefore, a global indicator is established to incorporate the drought problems derived from the resources of both the Segura and the Tajo. The proportion of each of them is established depending on their ranges of variation. The range of variation in the indicator of the transfer is lower than that of the basin and both control for a similar volume of demand (540 hm$^3$ and 495 hm$^3$). Therefore, the following formula is proposed [51]:

$$\text{Global Indicator System} = \alpha \cdot \text{Transfer Indicator} + \beta \cdot \text{Basin Indicator} \tag{3}$$

where the coefficients are calculated according to:

$$\alpha = 1 - (\text{Transfer Indicator Range/Total Range}) \tag{4}$$

$$\beta = 1 - (\text{Basin Indicator Range/Total Range}) \tag{5}$$

where the range is the difference between the maximum and minimum of the historical series for each indicator and the total the sum of the two.

After calculating the indicators, the drought thresholds are established through the Status Index (Ie): a dimensionless value between 0 and 1. The thresholds of the different drought statuses are related to the different degrees of satisfaction of the demands of the different uses, with the following thresholds (Table 5).

**Table 5.** Status indices and threshold values. Source: SDP 2007 [51].

| Status Indices | Threshold Values |
|---|---|
| Normality | Between 1 and 0.5 |
| Pre-alert | Between 0.5 and 0.35 |
| Alert | Between 0.35 and 0.2 |
| Emergency | Less than 0.2 |

As established by the SDP, once the status indices cross the limit of normality, the action measures will be activated. Finally, drought is understood as the situation when one or more of the previously defined drought indicators drop below the pre-alert level [51].

*3.3. Methodology SDP 2018*

As previously indicated, the SDP 2018 differentiates between the prolonged drought and temporary scarcity scenarios. The former is related to the reduction in rainfall and the contributions and the latter to the problem of meeting socioeconomic demands.

A priori, the territorial units for managing the two scenarios should be different. In a situation of prolonged drought, they are homogeneous in terms of resources (territorial units of drought, hereafter, TUD), and in the case of temporary scarcity in terms of demands and infrastructures (territorial units of scarcity, hereafter, TUS). However, in the SRB, the two types of territorial units for the analysis of prolonged drought and temporary scarcity are interrelated (Table 6).

**Table 6.** Relationship between TUD and TUS. Source: SDP 2018 [34].

| TUD | TUS |
|---|---|
| TUD 1—Principal System | TUS 1—Principal System |
| TUD 2—Headwaters Segura and Mundo | TUS 2—Headwaters Segura and Mundo |
| TUD 3—Left Bank Tributaries | TUS 3—Left Bank Tributaries |
| TUD 4—Right Bank Tributaries | TUS 4—Right Bank Tributaries |

As a basis for defining the TUD and TUS (Figure 4), in the SDP 2018, the hydraulic sub-zones defined in the Segura Basin Hydrological Plan of the second planning cycle 2015/21 (hereafter, SBHP 2015/21) have been taken as a reference [59], based on hydrographic, environmental, administrative and socioeconomic criteria, with hydrographic aspects taking preference. The indicator systems developed with the methodology of the SDP 2018 is based on these territorial units for the analysis of prolonged drought and temporary scarcity, defined in the SDP 2018 [34]:

(1) TUD-TUS 1 or Principal System: Dominated by the headwater reservoirs of the Talave, Fuensanta and Cenajo and the infrastructure of the distribution channels. In these

areas, the majority of the surface and groundwater resources of the basin are applied, together with the treated resources and all of the resources of the Tajo-Segura, those of the Negratín and the desalinated resources. Most of the population and the irrigated area of the basin are concentrated in this area, as is the application deficit (due to the lack of guarantee of the TTS resources), which partly represents structural scarcity. On the other hand, the over-exploitation of the groundwater resources (extractions exceeding renewable resources) amounts to 125 hm³/year. Finally, within this principal system there are three irrigation sub-systems with common characteristics (origin of resources, irrigation techniques and history): the Vegas del Segura; the irrigated areas of the transfer (hereafter, IAT); and the area outside of the IATs.

(2) TUD-TUS 2 or the Headwater System of the Segura and Mundo rivers: Waters above the Cenajo and Talave reservoirs. Practically all of the resources are surface resources of the river or springs.

(3) TUD-TUS 3 or the Left Bank Tributaries: Comprises the basins in the south-east of Albacete and the highlands of Murcia. The infrastructures are insufficient to apply the transferred or desalinated resources or those of the Segura River. The resources used are practically all groundwater resources, with the problem of the over-exploitation of the aquifers (100 hm³/year).

(4) TUD-TUS 4 or the Right Bank Tributaries: Comprises the basins discharging into the rivers Moratalla, Argos, Quípar and the Puentes reservoir. They are supplied by surface and groundwater and a substantial contribution of the springs in the area.

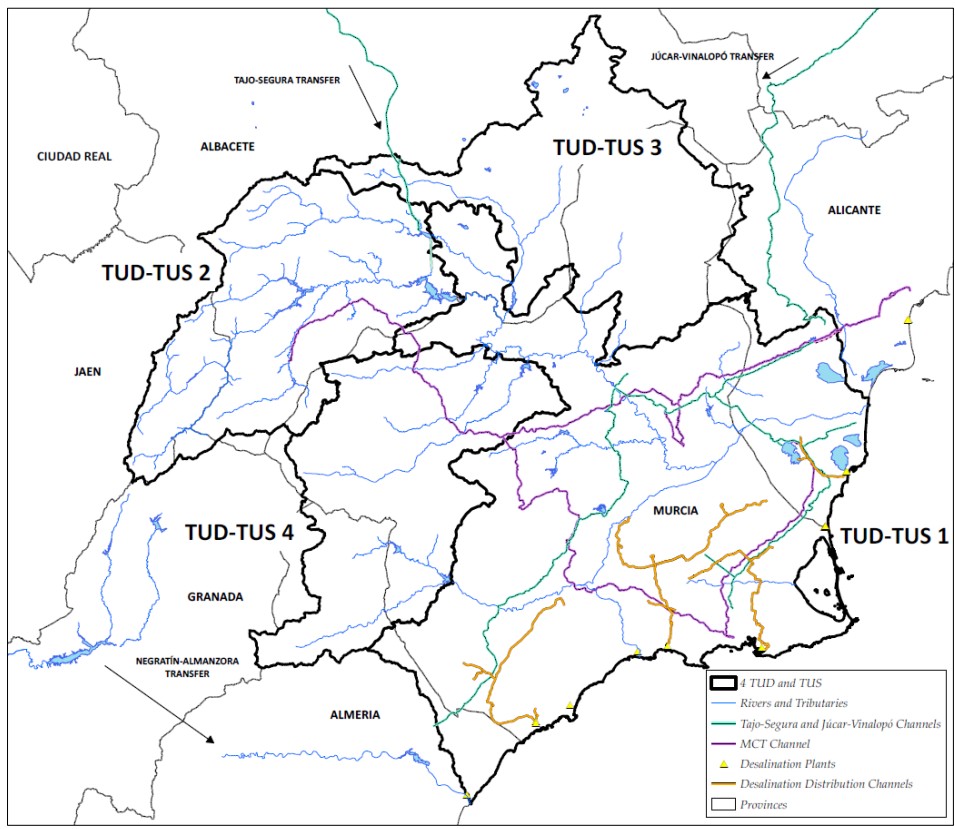

**Figure 4.** Territorial Unit in the SRB. Source: own elaboration based on [34].

Prolonged Drought:

The indicator selected in the SDP 2018, in each TUD, for the representation and analysis of prolonged drought is the 9-month SPI index (standardised rainfall index for a 9-month accumulation period) [60]. The nine-month SPI index is an indication of the inter-seasonal rainfall patterns in average time scales. For these time scales, low SPI values are

considered a good indication that the drought is having a significant impact on agriculture and may also be affecting other sectors [18].

Subsequently, for each of the TUDs, the Status Index has been calculated (Ie) through the standardisation of this index [34]:

- Maximum SPI = 1.00
- Median SPI = 0.50
- 10th percentile SPI = 0.30 Prolonged Drought (confirmation of fulfilment of ecological flow)
- Minimum SPI = 0.00

For the headwater TUDs, left bank tributaries and right bank tributaries, the Status Index applied corresponds to that of each TUD, based on the fulfilment of the ecological flows of their water bodies. However, for the principal TUD 1, and due to the influence exercised by the headwater system, the Drought Status Index is corrected with the Status Index of the headwater system in the main flow of the Segura River, due to the temporary deterioration of the water bodies and the relaxation of environmental flows [34], as we can observe in the following table (Table 7).

**Table 7.** Status indices of each TUD. Source: SPD 2018 [34].

| TUD | Status Indices Prolonged Drought | Length of River Bodies with Ecological Flow (km) | Weighting Factor | Weighting Factor |
|---|---|---|---|---|
| TUD 1 –Principal System | Status indices TUD 1 | 192.99 | 16.5% | **16.5%** |
| | Status indices TUD 2 | 194.34 | 16.6% | **60.2%** |
| TUD 2—Headwater | Status indices TUD 2 | 511.82 | 43.7% | |
| TUD 3—Left Bank Tributaries | Status indices TUD 3 | 10.72 | 0.9% | **0.9%** |
| TUD 4—Right Bank Tributaries | Status indices TUD 4 | 262.5 | 22.4% | **22.4%** |
| **TOTAL** | | **1172.37** | **100%** | **100%** |

In order to establish the Global Status Index of the basin, a weighting of the status indices of each TUD has been used, with the ecological flows based on the kilometres of water bodies of the river category. This Global Index is considered when declaring a prolonged drought. The values obtained for each of the TUDs are the following: headwater TUD 2: 60.2%; principal TUD 1: 16.5%; TUD 4 right bank tributaries: 22.4%; and TUD 3 left bank tributaries 0.9%. In this way, the Global Status Index of the basin (*Ie*) is established as [34]:

$$I_e = I_e^{Headwater} \cdot 0.602 + I_e^{Principal} \cdot 0.165 + I_e^{Right} \cdot 0.224 + I_e^{Left} \cdot 0.009 \qquad (7)$$

However, in the Segura Basin, a prolonged drought can also be declared if in the Tajo Basin a prolonged drought is declared in the TUD of the Tajo headwater (*Ie* < 0.3), in accordance with the contributions of the Entrepeñas and Buendía reservoirs (headwater in the Tajo Basin). To calculate this index, in the SDP of the Tajo, a weighting of the accumulated contributions (3 months) in the Entrepeñas and Buendía reservoirs is used through the following expression:

$$I_e = 0.55 \cdot Contributions\ Entrepeñas\ reservoir + 0.45 \\ \cdot Contributions\ Buendía\ reservoir \qquad (8)$$

Structural and Temporary Scarcity:

In the SDP 2018, two types of scarcity are defined. On the one hand, structural scarcity is defined *as a situation of continued scarcity which makes it impossible to fulfil the guarantee criteria with respect to the demands acknowledged in the corresponding hydrological plan.* On the other hand, temporary scarcity is defined *as a situation of non-continuous scarcity which, even enabling the fulfilment of the guarantee criteria in terms of meeting the demands established in the corresponding hydrological plan, temporarily limits the supply in a significant way.*

A solution to the structural scarcity problem should be provided in the next Segura Basin Hydrological Plan of the third cycle 2022/27. However, its value has been calculated in the SDP 2018. In order to better understand this problem of the SRB, it is necessary to first summarise the principal water demands and the deficit in meeting these demands, established in the SBHP 2015/21 [61].

Table 8 shows, for each type of demand, the average application of resources to meet gross demand, and which of these resources correspond to non-renewable withdrawals (over-exploitation of the aquifers). Finally, two deficits are identified. The application deficit, calculated as the difference between the demand and the resources applied, and the total deficit, calculated as the sum of the application deficit and the non-renewable withdrawals.

**Table 8.** Deficit SBHP 2015/21, on the 2015 horizon. Source: own elaboration based on [34,61].

| Demands | Applied Water (hm³/Year) | Gross Demand (hm³/Year) | Non-renewable Withdrawals (hm³/Year) | Application Deficit (hm³/Year) | Total Deficit (hm³/Year) |
|---|---|---|---|---|---|
| Agricultural | 1342 | 1546 | 226 | 203 | 429 |
| Urban | 236 | 236 | - | - | - |
| Unconnected industrial | 9 | 9 | 2 | - | 2 |
| Irrigation of golf courses | 11 | 11 | 3 | - | 3 |
| Environmental (wetlands) | 32 | 32 | - | - | - |
| Environmental (coastal aquifers) | 7 | 7 | - | - | - |
| **TOTAL** | **1637** | **1841** | **231** | **203** | **434** |

Due to the importance of irrigation in the SRB, (82% of the applied resources and 84% of the demand), Table 9 shows a summary of the data associated with this sector, which includes the breakdown of the gross and net areas of the 64 units of agricultural demand (hereafter UAD), the gross demand related to these UADs, the water applied, the application deficit and over-exploitation (non-renewable withdrawals, hereafter, NRW), for the 2015 horizon.

**Table 9.** Distribution by origin of the water for agricultural demand by TU (horizon 2015). Source: own elaboration based on [34,61].

| Territorial Unit (nº UAD) | Gross Area (ha) | Net Area [1] (ha) | Gross Demand (hm³/Year) | Applied Water (hm³/Year) | Application Deficit (hm³/Year) | NRW (hm³/Year) |
|---|---|---|---|---|---|---|
| Plains (9) | 57,460 | 35,369 | 252 | 252 | 0 | 0 |
| Transfer areas (18) | 150,770 | 88,049 | 617 | 435 | 181 | 24 |
| Outside transfer areas (19) | 145,513 | 76,508 | 430 | 415 | 15 | 105 |
| TUD-TUS 1 (46) | 353,743 | 199,926 | 1299 | 1102 | 196 | 129 |
| TUD-TUS 2 (4) | 8961 | 3097 | 17 | 17 | 0 | 0 |
| TUD-TUS 3 (7) | 93,977 | 44,171 | 153 | 153 | 0 | 96 |
| TUD-TUS 4 (7) | 33,637 | 15,199 | 77 | 70 | 7 | 0 |
| **TOTAL (64 UADs)** | **490,318** | **262,393** | **1546** | **1342** | **203** | **226** |

[1] The net area refers only to the UADs located within the Segura River Basin, namely 44 UADs in the TU I and 62 UADs in total.

As already mentioned, the basin has an application deficit in agricultural demands of 203 hm³/year, concentrated in the principal TUS 1, generated by the lack of guarantee of resources transferred from the Tajo (for irrigation an average of 205 hm³/year has been transferred, series 1980/81–2011/12, in contrast with the forecasted maximum of 400

hm³/year [58]). Furthermore, the use of non-renewable groundwater resources has given rise to an over-exploitation of 226 hm³/year.

The guarantee criteria (Figure 5) that should be fulfilled in order to meet the demand are stated in the HPI [62]. The criteria for agricultural use would not be fulfilled when:

- The deficit in a year is higher than 50% of the annual demand, or;
- The deficit in two consecutive years is higher than 75% of the annual demand, or;
- The accumulated deficit over 10 consecutive years is higher than 100% of the annual demand.

In the case of urban use, non-compliance will arise when:

- The deficit in a month is higher than 10% of the corresponding monthly demand, or;
- In 10 consecutive years, the sum of the accumulated deficit is higher than 8% of annual demand.

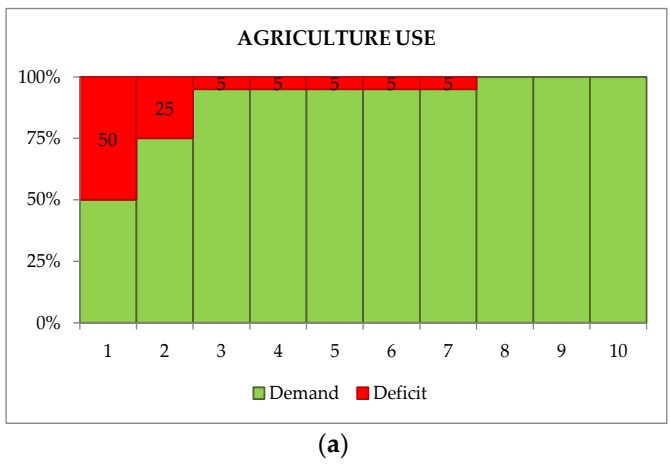
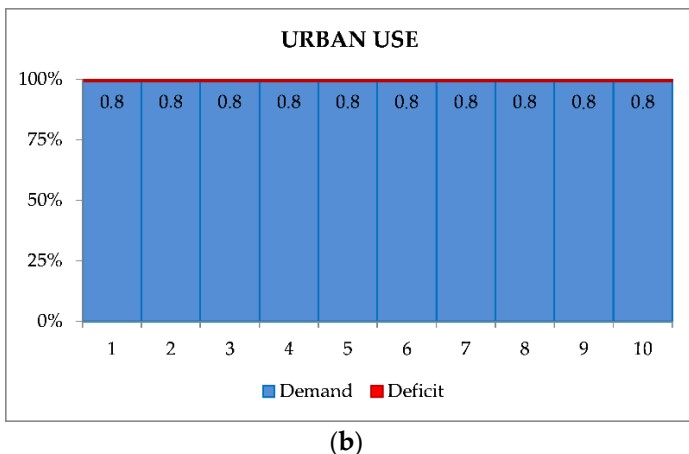

**Figure 5.** (**a**) Guarantee criteria agricultural use; (**b**) guarantee criteria urban use. Source: own elaboration based on [62].

To ensure that the global demand of the principal TUS 1 does not fail to comply with the guarantee criteria of the HPI (for agricultural use, as it does comply for urban use), and assuming that the non-renewable groundwater resources will be applied until 2027 at most, the resources from the Tajo-Segura transfer should exceed the 280 hm³/year for irrigation every year. Therefore, given that the average volume of transferred resources in the series 1980/81–2011/12 was 205 hm³/year in the destination for irrigation, the structural scarcity in the Segura Basin (Table 10) has been defined as 75 hm³/year [34].

**Table 10.** Structural scarcity on the SRB. Source: own elaboration based on [34].

| Territorial Unit (nº UAD) | Gross Demand (hm³/Year) | Application Deficit (hm³/Year) | NRW (hm³/Year) | Structural Scarcity (hm³/Year) |
|---|---|---|---|---|
| Plains (9) | 252 | 0 | 0 | 0 |
| Transfer areas (18) | 617 | 181 | 24 | 75 |
| Outside transfer areas (19) | 430 | 15 | 105 | 0 |
| TUD-TUS 1 (46) | 1299 | 196 | 129 | 75 |
| TUD-TUS 2 (4) | 17 | 0 | 0 | 0 |
| TUD-TUS 3 (7) | 153 | 0 | 96 | 0 |
| TUD-TUS 4 (7) | 77 | 7 | 0 | 0 |
| **TOTAL (64 UADs)** | **1546** | **203** | **226** | **75** |

With this minimum guaranteed volume of resources, a residual deficit of 20 hm³/year would still remain, but the guarantee criteria for the HPI will be met for the series of

demands of the principal TUS 1, the guarantee criteria of the HPI, including the current application of the non-renewable resources (129 hm³/year in the principal TUS).

However, in order to calculate the indicator of temporary scarcity, the relationship between the availability of resources and the demands has been established. The situations of temporary deficit in each of the defined TUS have been identified so as to obtain a single indicator of temporary scarcity for each TUS.

In the case of TUS 1 where a mixture of own and transferred resources is produced, we have considered the contributions accumulated over the previous 12 months in the headwaters of the Segura Basin and the reservoir-stored resources of the basin (basin resources indicator). At the same time, the contributions accumulated over the previous 12 months and the reservoir-stored resources of the Tajo Basin (transfer resources indicator) have also been calculated. In order to calculate the global indicator of TUS 1, a 50% distribution of the aforementioned indicators has been considered [34], as shown in the following formula:

$$\text{BASIN RESOURCES Indicator} = \frac{2 \cdot \text{Contributions} + \text{Stores resources}}{3} \qquad (10)$$

where the contributions are those accumulated over the previous 12 months and the stored resources those in the Fuensanta, Cenajo, Camarillas, Talave and Alfonso XIII reservoirs (Segura Basin) on the date of the calculation.

$$\text{TRANSFER RESOURCES Indicator} = \frac{\text{Contributions} + 2 \cdot \text{Stored resources}}{3} \qquad (11)$$

where the contributions are those accumulated over the previous 12 months in the headwaters of the Tajo, and the stored resources are those in the reservoirs of Entrepeñas and Buendía (Tajo Basin) on the date of calculation.

$$\text{GLOBAL Indicator} = 50\% \text{ Transfer Resources Indicator} + 50\% \text{ Basin Resources Indicator} \qquad (12)$$

With a scarce regulation of resources to meet their demands, the rest of the TUS fundamentally depend on meteorological drought. Therefore, the prolonged drought itself has been used as an indicator of temporary scarcity: nine-month SPI [34].

Table 11 shows a summary of the indicators selected.

**Table 11.** Indicator of temporary scarcity. Source: own elaboration based on [34].

| TUS | Indicator |
|---|---|
| TUS 1—Principal System | BASIN RESOURCES Indicator |
| | TRANSFER RESOURCES Indicator |
| | GLOBAL Indicator |
| TUS 2—Headwater | Nine-month SPI |
| TUS 3—Left Bank Tributaries | Nine-month SPI |
| TUS 4—Right Bank Tributaries | Nine-month SPI |

In order to calculate the Status Index (Ie) for each of the variables selected in each TUS, it is necessary to carry out a re-scaling (with values between 0 and 1) of the value of each indicator so as to enable a comparison to be made of the status of any TUS [34].

- A value of 0.50 of the index will correspond to the pre-alert threshold defined for the variable.
- A value of 0.30 of the index will correspond to the alert threshold defined for the variable.
- A value of 0.15 of the index will correspond to the emergency threshold defined for the variable.

Due to the weight of the demands of the TUS 1 in relation to the whole basin (84%) and given that this is the TUS with an under-resourcing problem due to the lack of

guarantee of the Tajo transfer, the SDP 2018 has established the scarcity indicator of the principal TUS as the scarcity indicator of the Global System (Table 12).

**Table 12.** Status indices and threshold values, temporary scarcity. Source: SDP 2018 [34].

| Status Indices | Threshold Values |
| --- | --- |
| Normality | Between 1 and 0.5 |
| Pre-alert | Between 0.5 and 0.3 |
| Alert | Between 0.3 and 0.15 |
| Emergency | Less than 0.15 |

When the indicator reaches a situation of pre-alert, the savings and demand control measures are automatically activated. In the alert scenario, as well as the previous measures, alternative resources are mobilised and supply restrictions may be contemplated. Finally, in emergency situations, exceptional and extraordinary measures are implemented in scenarios of severe scarcity [34].

Extraordinary Drought:

Finally, after defining prolonged drought in the Segura Basin (Equation (6)) and in the Tajo Basin (Equation (7)) and temporary scarcity (Equation (10)), as indicated in the SDP 2018, the President of the Segura Hydrographic Confederation is able to declare an exceptional situation due to extraordinary drought when in the whole of the basin there are:

- Alert scarcity scenarios that temporarily coincide with that of a prolonged drought (either in the Segura or Tajo Basin).
- Emergency scarcity scenarios.

In this exceptional situation due to extraordinary drought and for the area affected by the declaration, the governing body of the basin organisation will assess the need and timeliness of requesting the government, through the ministry responsible for water, to adopt measures related to the use of the public hydraulic domain, as established in Article 58 of the Rewritten Text of the Water Act (TRLA) [34].

## 4. Results

After analysing the methodologies established in the SDP 2007 and SDP 2018 for identifying and declaring droughts, the following sections present the results obtained by applying both methodologies to the period of study (2015–2019).

### 4.1. Characterisation of the Drought of 2015–2019 with the SDP 2007

As previously indicated, the methodology used in the SDP 2007 is based on calculating three status indices: one for the basin system, another for the transfer system and a Global Index. The following graphs show the evolution of the three indices for the period between October 2010 and December 2019.

The first graph (Figure 6) shows the evolution of the index of the Basin System, which displays a decreasing trend from the beginning of 2014. The pre-alert threshold was reached in June 2016, the alert level in May 2017 and the emergency level in the period between September 2017 and February 2018. Finally, we should point out that on the 1st of May 2015, the date when the drought was declared, the Status Index of the Basin System was in a situation of normality.

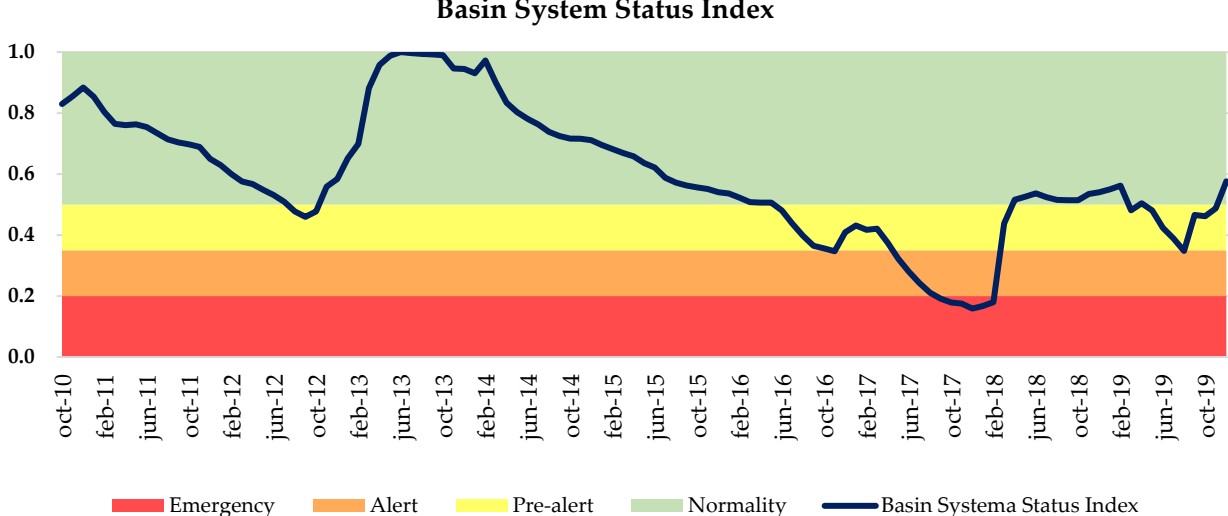

**Figure 6.** Evolution of the Basin System Status Index, period October 2010-December 2019. Source: own elaboration based on [62].

The second graph (Figure 7) shows the evolution of the Transfer System Index, where we can observe a more unfavourable situation as it displays a decreasing trend from March 2014 starting at a value of 0.661, reaching the pre-alert level in July 2014, the alert level in January 2015 and the emergency level in two long periods; July 2015–February 2016 and January 2017–March 2018. Finally, we should point out that on the 1st of May 2015, the date when the drought was declared, the Status Index of the Transfer System was in a situation of alert.

This scenario was particularly severe in the irrigated areas linked to the Tajo-Segura transfer, where the users suffered a reduction in the available resources, from 142.5 hm³, which was the agreed volume for irrigation to be transferred between October and February 2014, to 94,5 hm³ in the water year 2014/15 [52].

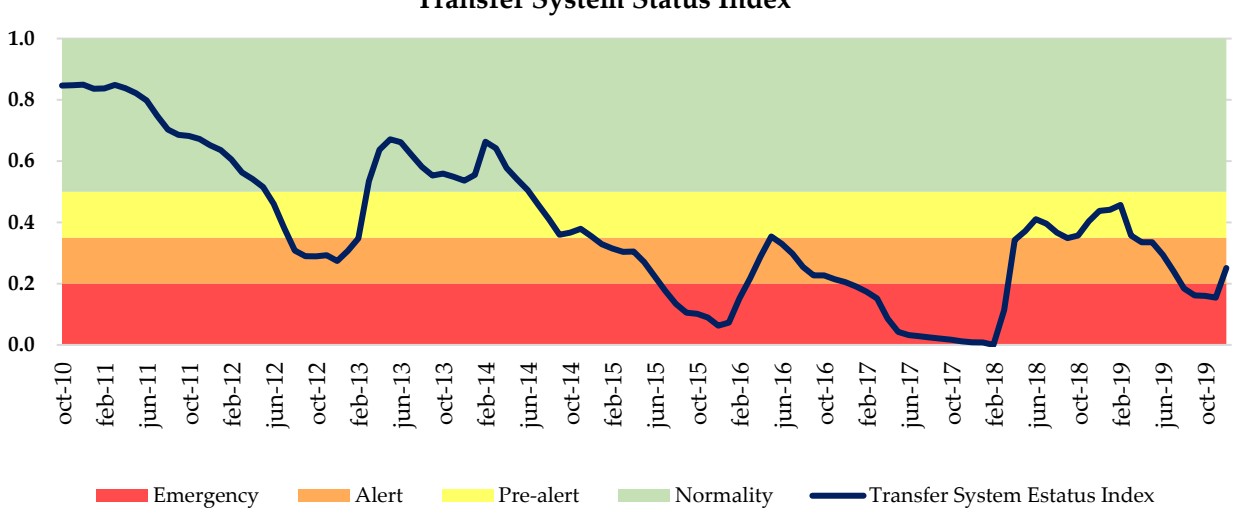

**Figure 7.** Evolution of the Transfer System Status Index, period October 2010-December 2019. Source: own elaboration based on [63].

The third figure (Figure 8) shows the evolution of the Global Status Index, where we can observe how the trend also clearly decreased from March 2014 (starting at a value of 0.891), reaching a pre-alert level in February 2015, an alert situation in July 2015 and an

emergency situation in two periods: the first between December 2015 and January 2016 and then, and more significantly, between April 2017 and March 2018. We should point out that on the 1st of May 2015, the date when the drought was declared, the Global Status Index was in a situation of pre-alert.

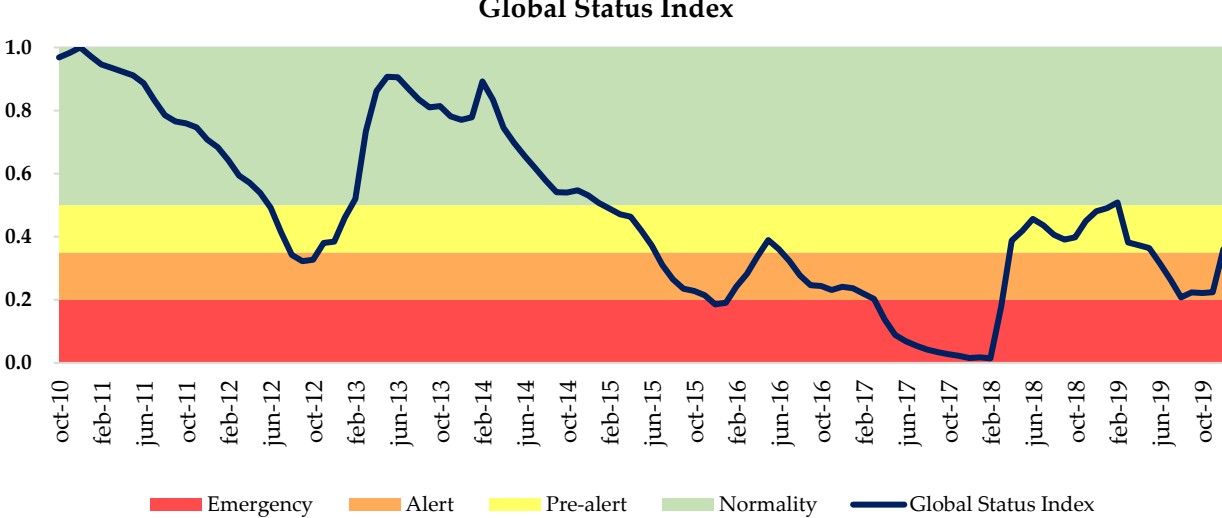

**Figure 8.** Evolution of the Global Status Index, period October 2010-December 2019. Source: own elaboration based on [63].

Therefore, after analysing the three drought indices defined in the SDP 2007, we can indicate that, on the 1st of May 2015, when the drought was declared, there was a pre-emergency situation in the Global Index and in the Transfer System Index and one of normality in the Basin System Index. However, the clearly descending trend in the three indices triggered the declaration of the drought and not the situation or alert or emergency in any of them.

*4.2. Characterisation of the Drought of 2015–2019 with the SDP 2018*

As previously indicated, the methodology used in the SDP 2018 differentiates between the situations of prolonged drought and temporary scarcity.

The following graph (Figure 9) represents the evolution of the prolonged drought indices of the Segura Basin and the headwaters of the Tajo Basin in the period between October 2010 and December 2019. In the case of the Segura Basin, after the declaration of drought in May 2015, there was a situation of prolonged drought between October 2015 and January 2016. However, there was prolonged drought in the headwaters of the Tajo Basin in two long periods: July 2015 to January 2016 and October 2016 to February 2018.

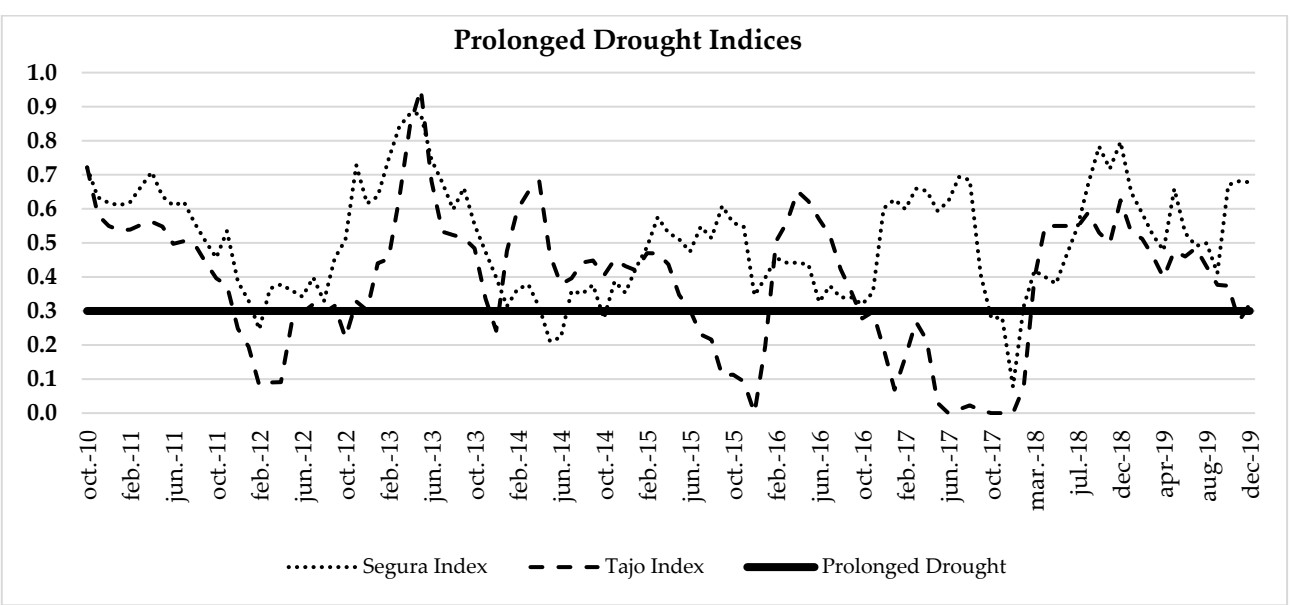

**Figure 9.** Evolution of the Prolonged Drought Indices of the Segura Basin and the headwaters of the Tajo Basin, period October 2010–December 2019. Source: own elaboration based on [64].

With respect to the indices of temporary scarcity, the SDP 2018 establishes that for the TUS of the headwaters and the left and right tributaries, the scarcity indices are also the 9-month SPIs. In contrast, the Principal TUS System Index, considered as the Global Index for the whole basin, is established as the average of the indices of the own resources of the basin and the resources of the Tajo-Segura transfer. The three temporary scarcity indices, according to the methodology of the SDP 2018, are graphically represented in the following figures.

The first graph (Figure 10) shows the evolution of the index of the basin resources, which displays a decreasing trend from March 2014 (value of 0.972). The pre-alert level was reached in July 2016, the alert level in June 2017 and the emergency level in the period between October 2017 and March 2018. Finally, we should point out that on the 1st of May 2015, the date when the drought was declared, the Status Index of the basin was in a situation of normality.

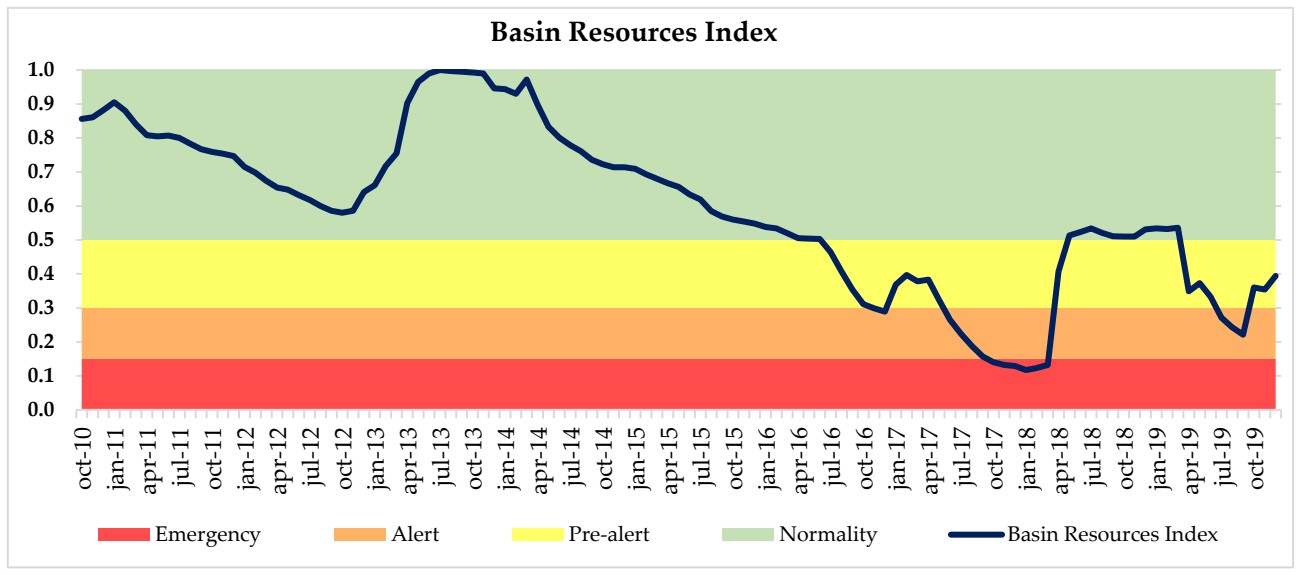

**Figure 10.** Evolution of the index of the basin resources (temporary scarcity) in the Segura Basin, period October 2010–December 2019. Source: own elaboration based on [64].

The second graph (Figure 11) shows the evolution of the index of the transfer resources, where we can observe a more unfavourable situation as it displays a decreasing trend from March 2014 (value of 0.686), reaching the pre-alert level in July 2014, the alert level in June 2015 and the emergency level in two long periods: August 2015–March 2016 and March 2017–April 2018. Finally, we should point out that on the 1st of May 2015, the date when the drought was declared, the Status Index of the Transfer System was in a situation of pre-alert and very close to a situation of alert.

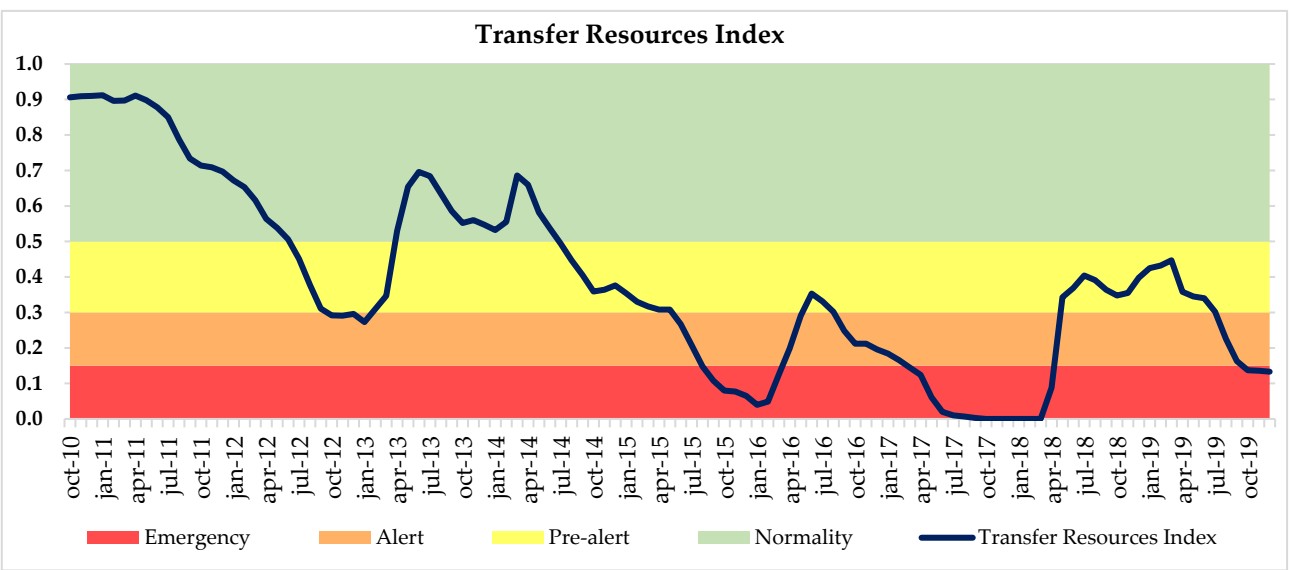

**Figure 11.** Evolution of the index of the transfer resources (temporary scarcity) in the Segura Basin, period October 2010–December 2019. Source: own elaboration based on [64].

The third figure (Figure 12) shows the evolution of the Global Index, where we can observe how the trend also clearly decreased from March 2014 (starting at a value of 0.829), reaching a pre-alert situation in March 2015, an alert situation in January 2016 and an emergency situation in just one period between June 2017 and March 2018. We should point out that on the 1st of May 2015, the date when the drought was declared, the Global Status Index was in a situation of pre-alert.

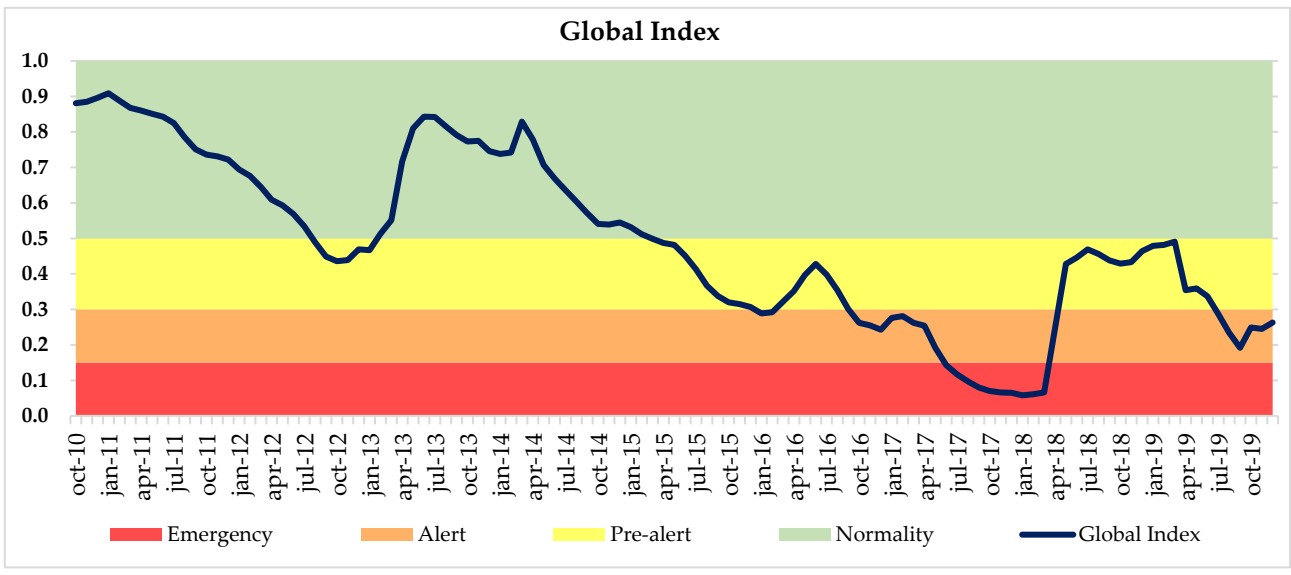

**Figure 12.** Evolution of the Global Index (temporary scarcity) in the Segura Basin, period October 2010–December 2019. Source: own elaboration based on [64].

Finally, and as previously mentioned, the SDP 2018 indicates that the President of the Segura Hydrographic Confederation is able to declare an exceptional situation due to extraordinary drought when the whole of the basin displays alert scenarios of scarcity coinciding in time with the prolonged drought (either in the Segura or Tajo Basin) or emergency scenarios of scarcity.

As we can observe in the following graph (Figure 13), with the new methodology of the SDP 2018, in the current drought period (2015–2019) the situation of extraordinary drought could be declared in two periods: the first in January 2016 (one month) and the second between October 2016 and April 2018 (19 months).

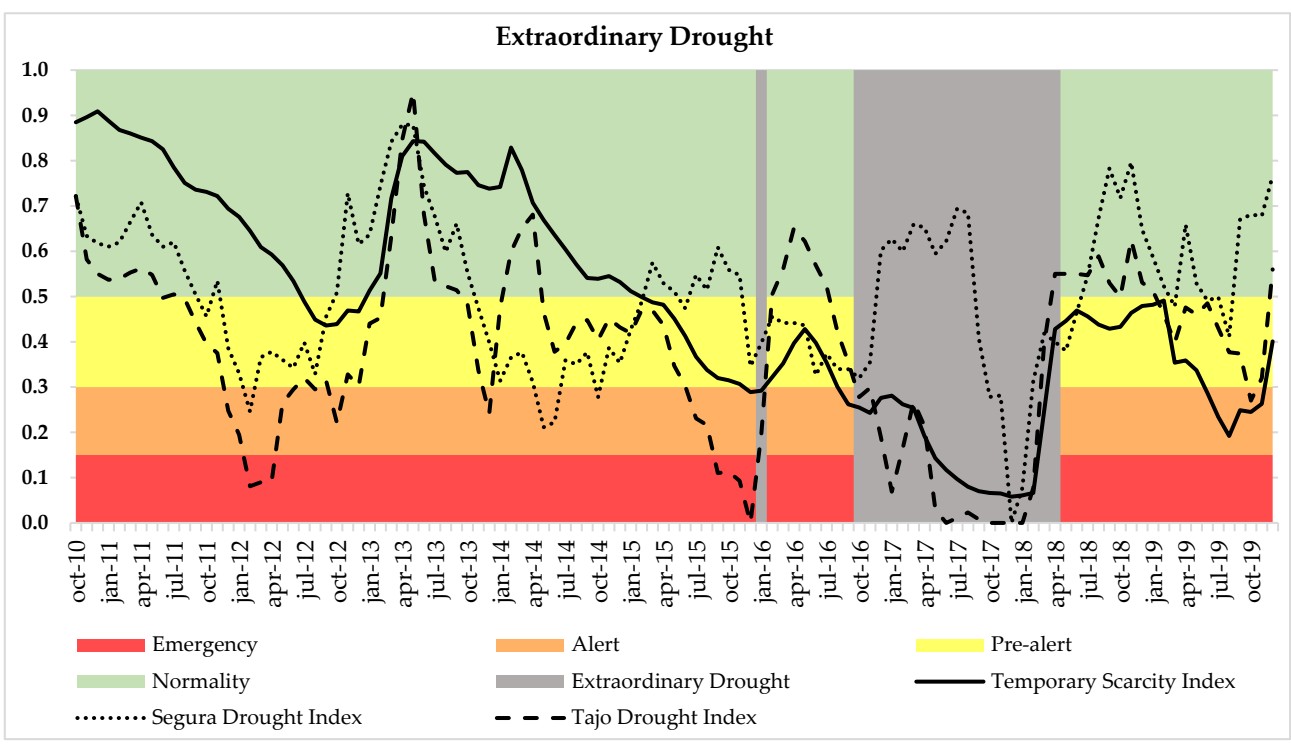

**Figure 13.** Evolution of the extraordinary drought in the Segura Basin, period October 2010- December 2019. Source: own elaboration based on [64].

## 5. Discussion

In January 2015, the basin system indicator was in a situation of normality (0.695), while the transfer system indicator was in a situation of pre-alert (0.329), which enabled the draft of the Drought Decree to be elaborated.

The aforementioned decree places emphasis on the fact that the decrease experienced from the beginning of 2014 was due to the reduction in the inter-annual contributions generated in the headwaters of the Segura and the Tajo. The inter-annual contribution (that of the previous 365 days) in the Segura Basin in March 2014 was 752 hm³, and in March 2015 it was just 381 hm³, representing a very sharp reduction of 50% [52].

This situation aggravated the existing deficit of resources in the Segura Basin which, with normal rainfall, would be 480 hm³/year. This was associated with the over-exploitation of groundwater and the under-watering of the existing crops, principally in irrigated areas of the Tajo-Segura transfer. This hindered the fulfilment of the environmental objectives for the different water bodies within the deadlines defined in the Segura Basin Hydrological Plan [51].

The conditions for the activation of the Drought Decree arose when the transfer system indicator entered into a situation of alert (0.325) in January 2015 but it was not until May 2015 when a drought in the SRB was declared (through Royal Decree 356/2015 of 8 May) based on the drought indicators established in the SDP 2007: the basin system

indicator was in a situation of normality (0.636), the transfer system indicator in a situation of alert (0.267) and the global system indicator in a situation of pre-alert (0.416).

The passing of the Drought Decree enabled the implementation of exceptional administrative measures in the management of water resources so as to mitigate the situation of scarcity, such as the authorisation of assignment of rights agreements, the commissioning and execution of drilling, the mobilisation of new resources from seawater desalination, the use of reservoir water to defend against floods and the desalobration of groundwater.

The Royal Decree Law 6/2015 of 14 May was also approved in May 2015, which modified Law 55/2007 of 28 December [65], granting extraordinary credit to address the needs derived from the drought situation in the SRB for a total amount of EUR 30 million, and an exception and temporary rule was approved regarding the transfer of rights to the exclusive use of the water from the SRB.

In addition, and with the backing of the regulations approved, other actions were implemented that enabled the mobilisation of extraordinary resources, as shown in Table 13.

**Table 13.** Other actions carried out in the area of the SRB during the water year 2014/15. Source: own elaboration based on [52,65,66].

| Date | Document/Information | Actions |
|------|---------------------|---------|
| June 2015 | Public information request from SCRATS (Aqueduct Central Irrigation Union) | Use of 9.6 hm³ from the Sinclinal de Calasparra aquifer and 15 hm³ of non-assigned resources of the Pedrera reservoir |
| | | Request for authorisation for the supply of 35 hm³ (non-assigned resources of the Pedrera) |
| | Approval of the supply by the SHC of resources from the Judió and Cárcabo retention reservoirs | Irrigation communities of Mazarrón, Margen Derecha Pilar de la Horadada, Águilas, Murada Norte, Fuente Librilla and the Sociedad Civil Virgen del Rosario (2 hm³) |
| July 2015 | MAGRAMA through the SHC | Main pipeline of desalinated water from Águilas to Valle Guadalentín (Lorca and Totana) |
| | Awarding of emergency works RD-Law 6/2015 | 15 actions to date |
| | | Temporary transfer of rights from the Irrigation Communities of Poveda and Canal Estremera (Region of Madrid) to the SCRATS |
| | MAGRAMA | Conditioning of the Sinclinal de Calasparra aquifer battery of wells |
| | RD 356/2015 Actions | Implementation of external wells up to a volume of 6 hm³. Authorisation for the extraction from two wells (extraction 0.48 hm³) |
| August 2015 | Approval by the Council of Ministers | Implementation of strategic battery drilling (SBD) in the Vega Media and Sinclinal de Calasparra aquifers |
| | | Authorisation of existing wells to the Trasvase Tajo-Segura Calasparra-Cieza Water User Association |
| | | Authorisation of Norte de la Vega del Río Segura UA to use the wells of Sinclinal de Calasparra and those of El Molar aquifers |

Notes: SCRATS = Tajo Segura Aqueduct Central Irrigation Union, WUA = Water User Association, WUAs = Water User Associations, MAGRAMA = Ministry of Agriculture, Food and the Environment, SBD = strategic battery drilling, UA = Users Assembly.

The evolution of the indicators continued to decline considerably, particularly prominent in the transfer system indicator. On 1 September 2015, the basin system indicator

was in a situation of normality (0.562), the transfer system indicator in a situation of emergency (0.105) and the global system indicator in a situation of alert (0.235).

This worsened situation led to the extension of the Drought Decree until 30 September 2016, through the passing of Royal Decree 817/2015 of 11 September, which contemplated additional measures to mitigate the effects of the drought. While the first extension of the Drought Decree was in force, the following actions were implemented in the SRB (Table 14).

**Table 14.** Actions carried out in the area of the SRB during the water year 2015/16. Source: own elaboration based on [53,66].

| Date | Document/Information | Actions |
|---|---|---|
| October 2015 | New measures announced by the Ministry | Price reduced €0.30/m³ desalinated water Torrevieja (30 hm³ and €6M) |
| | | Subsidies of €0.10/m³ desalinated water Valdelentisco (20 hm³ and €2M) |
| | | General budgets 2016: Heightening of Camarillas dam and two new dams in Lébor and Moreras |
| January 2016 | Council of Ministers | Conditioning and exploitation of Sinclinal de Calasparra aquifer wells (to date 30.7 hm³ in 2015) |
| February 2016 | Council of Ministers | Execution of works on the Sinclinal de Calasparra aquifer wells |
| | | Emergency works on the El Molar aquifer wells |
| | | Emergency works, execution of tasks to monitor use and hydrological information |
| March 2016 | MAGRAMA | Termination of pipeline—Águilas-Valle Guadalentín (27 km and €20M), 150,000 m³/day |
| April 2016 | MAGRAMA, ACUAMED | Agreement with the Mazarrón Water User Association regarding Valdelentisco resources |

Notes: WUA = Water User Association, MAGRAMA = Ministry of Agriculture, Food and the Environment, ACUAMED = Waters of the Mediterranean Basins.

On 1 September 2016, the indicators continued to register low values, close to the emergency level. The basin system was in a situation of pre-alert (0.365) as a result of the lack of rainfall during the water year 2015/16; the transfer system was in a situation of alert (0.227), moderately recovering from February 2016 due to the increase in rainfall in the headwaters of the River Tajo; and the global indicator was in a situation of alert (0.246), which led to the passing of a new extension of the Drought Decree until 30 September 2017, through the approval of Royal Decree 335/2016, of 23 September [54].

During the month of June 2017, the Royal Decree-Law 10/2017 of 9 June was passed, referring to urgent measures to mitigate the effects of drought (in the basins of the Segura, Júcar and Duero rivers). Of the actions carried out, we can highlight the exemptions from the regulation rate and the tariff quotas (€37 M savings) for the holders of the water use rights for irrigation and for the MCT, together with moratoriums on Social Security contributions [67].

The actions implemented during the period when the second extension of the Drought Decree was in force are shown in the following table (Table 15).

**Table 15.** Actions carried out in the area of the SRB during the water year 2016/17. Source: own elaboration based on [54,67].

| Date | Document/Information | Actions |
|---|---|---|
| September 2016 | Council of Ministers | Emergency pipeline works desalination plant of Valdelentisco-Algeciras reservoir |
| March 2017 | MAPAMA | The SHC awards the users of the SCRATS 21 hm³ from the Torrevieja desalination plant |
| | | Temporary transfer of desalinated water from desalination plant San Pedro to the SCRATS (0.5 hm³/month and 2 months) |
| June 2017 | New drought measures | Opening of Sinclinal Calasparra aquifer wells (31.9 hm³) |
| | | Opening of Vega Alta aquifer wells (4.5 hm³) |
| July 2017 | Public Information Process, EIA of Campo Cartagena aquifer | Opening of 252 wells and the extraction of 28.6 hm³ |
| August 2017 | MAPAMA | MCT Desalination plant (Alicante I and II, San Pedro) and ACUAMED (Torrevieja, Valdelentisco and Águilas) |
| | Emergency works to increase the performance of the desalination plants in the MCT | Beginning of works on MCT desalination plants in Alicante (€2.3M) |
| | Production of desalinated resources, ACUAMED | To date in 2017: 75.7 hm³. Valdelentisco 21 hm³, Águilas 29.5 hm³ and Torrevieja 25.2 hm³ |
| | MAPAMA through the ACUAMED Additional resources | From the desalination plant of Águilas to the irrigation lands of the coastal area of Águilas and Pulpí and the Valle Guadalentín |
| | MAPAMA through the SHC implementation of the battery of wells | Implementation of seven of the 15 wells of the Vega Media aquifer (for this month 3.5 hm³) |

Notes: SCRATS = Tajo Segura Aqueduct Central Irrigation Union, MAPAMA = Ministry of Agriculture, Fishing and Food, MCT = Mancomunidad de los Canales de Taibilla, management entity of water distribution, ACUAMED = Waters of the Mediterranean Basins, EIA = Environmental Impact Assessment.

The lack of rainfall in these last three years in the headwaters of the Segura and Tajo and particularly in the last year, 2017, led to the decrease in the contribution to the reservoirs and the volume stored in them. Within this context, the contribution received by the reservoirs of the headwaters of the Segura between June 2016 and 2017 was 222 hm3, which is less than 70% of the historical average of the last 30 years. In September 2017, the indicators were at minimum levels and in a situation of emergency (the basin system (0.191), the transfer system (0.021) and the global system (0.034)). This led to the passing of the third extension of the Drought Decree until 30 September 2018, through the approval of Royal Decree 851/2017 of 22 September [55].

During the time when the third extension of the Drought Decree was in force, actions were approved by the Council of Ministers such as the increase in desalinated water (private desalination plants and ACUAMED) and the incorporation of new resources derived from the conditioning and recovery of the Segura siphon [66].

In December 2017, the three-month public consultation period began of the review of the Special Drought Plan (which was eventually approved in 2018), through its publication on 21 December in the Official State Gazette.

One of the most important milestones of the year 2018 in addressing the drought situation was the passing of Law 1/2018 of 6 March referring to urgent measures to mitigate the effects generated by the drought. Some of the most relevant measures were [68]:

- Employment and Social Security improvements.
- Special tax reductions for agricultural activities.

- Application for an advance of the subsidies of the Common Agricultural Policy (CAP) and the financing of guarantees.
- Aid for the combined agricultural insurance plan.
- ICO measures loans (Official Credit Institute).
- Creation of an Extraordinary Fund to combat drought, which for the year 2017 amounted to €1000 M.
- Modification of the types of taxes (which required the modification of the Revised Text of the Water Law).
- Exemptions related to the availability of water (rate charge for water use and regulation charge, fixed and variable costs of the water conveyance rate and Tajo-Segura post-transfer conveyance rates).

With respect to this last point, it is important to indicate the tariffs and charges, prior to the passing of Law 1/2018, referring to surface water services, corresponding to the annual volumes captured or derived from surface water bodies through public services (volumes discharged from the reservoirs and transported by the principal infrastructures to the areas of downstream supply) [39]. They are reflected in Table 16.

**Table 16.** Tariffs and charges used in the cost recovery analysis for use in irrigation. Source: own elaboration based on [39].

| SERVICE | TARIFF APPLIED | | VALUE | UNIT |
|---|---|---|---|---|
| **Surface water services** | TTS water transferred | | 0.117893 | €/m³ |
| | Own TTS resources | | 0.023178 | €/m³ |
| | Regulation charge of the Segura, Mundo and Quipar rivers | Irrigation charge prior to 1933 | 13.95 | €/ha |
| | | Irrigation charge after to 1933 | 17.31 | €/ha |
| | | Irrigated area of Hellín | 6.95 | €/ha |
| | Regulation charge of the River Mula | La Cierva WUA | 43.23 | €/ha |
| | | Purísima de Yéchar WUA | 56.80 | €/ha |
| | | Heredamiento Puebla de Mula | 42.08 | €/ha |
| | Regulation charge of the River Guadalentín | | 23.79 | €/ha |
| | Regulation charge of the River Argos | | 73.85 | €/ha |

During the water year 2017/18 and specifically from February, the situation of the indicators improved, which translated into a significant recovery, but was not sufficient to end the situation of drought. In September 2018, the basin system indicator was in a situation of normality (0.514), the transfer system indicator in a situation of alert (0.348) and the global system indicator in a situation of pre-alert (0.391). In accordance with these data, a fourth extension of the Drought Decree was approved until 30 September 2019, with the passing of Royal Decree 1210/2018 of 28 September [56].

In November 2018, the review of the Special Drought Plans (SDP 2018) corresponding to the intra-community basins, including the SRB, was approved through the Order TEC/1399/2018 of 28 November, which represented, as previously commented, a change in the methodology for calculating the drought and scarcity indicators and in the activation of the Drought Decree.

The fourth extension of the Drought Decree had a validity of one year and ended on 30 September 2019. Even though a situation of normality had not been recovered, the status of the indicators did not allow for the approval of a new extension. On 1 September 2019, the Prolonged Drought Index in the SRB was in a situation of normality (0.441) as was the Drought Index in the headwaters of the Tajo River (0.374). In this situation of an absence of drought, the SDP 2018 determines that in order to activate (or extend) the Drought Decree, the global scarcity indicator should be in a situation of emergency; however, it was in a situation of alert (0.192), as was the basin indicator (0.221) and the transfer indicator (0.163).

Therefore, the Drought Decree, with its four extensions, which were approved based on the indicators of the SDP 2007, was in force from May 2015 until September 2019: a total of 53 months. However, as analysed in the section of the characterisation of drought with the SDP 2018, the conditions for declaring an extraordinary drought arose in two periods, during the month of January 2016 and from October 2016 to May 2018 (19 months); a total of 20 months.

It is essential to identify this difference, as the application of extraordinary resources after the passing of the Drought Decree allows the available water to be increased, mobilising non-assigned groundwater resources [69], the increase of the production of desalinated water, together with other resources mobilised from the basin itself and some transfer agreements with irrigators of other basins [70,71]. This mitigates the reduction in resources and maintains the cultivated area of the agricultural sector (main destination of the water resources, accounting for more than 80% of the demand of the SRB) in a situation of normality, at least in the most productive areas of the SRB [42].

In the follow-up report of the natural year 2019, and water year (hereafter, WY) 2018/19 [70], and particularly in its Section 3.6 (extraordinary resources in accordance with RD 365/2015), the maximum authorised extraordinary resources were identified (this does not imply that they were mobilised) by the basin organisation pursuant to the Drought Decree, in order to mitigate the drought situation, in terms of both supply and irrigation, from the water year 2014/15, as shown in the following Table 17:

**Table 17.** Maximum authorised extraordinary resources (m³/year). Source: own elaboration based on [70,72].

|  | WY 2014/15 | WY 2015/16 | WY 2016/17 | WY 2017/18 | WY 2018/19 | Total (m³) |
|---|---|---|---|---|---|---|
| **Underground extractions** | 36,822,500 | 58,167,695 | 48,154,533 | 113,118,339 | 20,712,946 | **276,976,013** |
| **Desalination** | 40,583,625 | 5,347,500 | 56,434,500 | 49,970,000 | 108,660,000 | **260,995,625** |
| **Dams and other resources** | 55,987,793 | 2,000,000 | 3,860,000 | - | - | **61,847,793** |
| **Contracts for assigning water rights** | 9,100,000 * | 10,900,000 * | 8,900,000 * | 1,750,000 | - | **30,650,000** |
| **TOTAL** | **142,493,918** | **76,415,195** | **117,349,033** | **164,838,339** | **129,372,946** | **630,469,431** |

\* Melgarejo-Moreno, J. y López-Ortiz, M.I. (2018) [70].

In order to study the repercussions of providing extraordinary resources on the economic sector of agriculture, an analysis is made of the value of production ($€_{2016}$M/year) and net margin ($€_{2016}$M/year) associated to the UADs related to the water applied in each year of the drought [42]. Tables 18 and 19 show the results obtained for the five water years of the drought compared with the maximum and average values obtained in the SBHP 2015/21:

**Table 18.** Evolution of the value of production 2015–2019. Source: [42] based on Hydrological Planning Office of the Segura Hydrographic Confederation (HPO).

| Territorial Unit (nº UAD) | Maximum Production Value ($M€_{2016}$/Year) | Average Production Value ($M€_{2016}$/Year) | Production Value 2015 ($M€_{2016}$/Year) | Production Value 2016 ($M€_{2016}$/Year) | Production Value 2017 ($M€_{2016}$/Year) | Production Value 2018 ($M€_{2016}$/Year) | Production Value 2019 ($M€_{2016}$/Year) |
|---|---|---|---|---|---|---|---|
| TU I: Principal (44) | 2482 | 2339 | 2339 | 2293 | 2286 | 2352 | 2365 |
| TU II: Headwaters (4) | 29 | 29 | 25 | 24 | 25 | 24 | 25 |

| | | | | | | |
|---|---|---|---|---|---|---|
| TU III: Left Bank Tributaries (7) | 337 | 337 | 329 | 317 | 317 | 328 | 330 |
| TU IV: Right Bank Tributaries (7) | 156 | 153 | 137 | 128 | 134 | 134 | 134 |
| **TOTAL (62)** | **3003** | **2857** | **2830** | **2762** | **2761** | **2838** | **2854** |
| Outside UAD | 0 | 0 | 95 | 90 | 98 | 104 | 115 |
| **TOTAL** | **3003** | **2857** | **2926** | **2852** | **2859** | **2942** | **2969** |

**Table 19.** Evolution of net margin 2015–2019. Source: [42] based on HPO.

| Territorial Unit (nº UAD) | Maximum Net Margin (M€$_{2016}$/Year) | Average Net Margin (M€$_{2016}$/Year) | Net Margin 2015 (M€$_{2016}$/Year) | Net Margin 2016 (M€$_{2016}$/Year) | Net Margin 2017 (M€$_{2016}$/Year) | Net Margin 2018 (M€$_{2016}$/Year) | Net Margin 2019 (M€$_{2016}$/Year) |
|---|---|---|---|---|---|---|---|
| TU I: Principal (44) | 1136 | 1091 | 1085 | 1068 | 1064 | 1091 | 1097 |
| TU II: Headwaters (4) | 12 | 12 | 10 | 10 | 10 | 10 | 10 |
| TU III: Left Bank Tributaries (7) | 154 | 154 | 151 | 146 | 146 | 150 | 151 |
| TU IV: Right Bank Tributaries (7) | 71 | 70 | 63 | 59 | 61 | 62 | 61 |
| **TOTAL (62)** | **1373** | **1326** | **1309** | **1283** | **1282** | **1313** | **1320** |
| Outside UAD | 0 | 0 | 44 | 42 | 46 | 48 | 53 |
| **TOTAL** | **1373** | **1326** | **1353** | **1324** | **1327** | **1361** | **1373** |

As we can observe in Tables 18 and 19, after applying the extraordinary resources mobilised through the activation of the measures established with the Drought Decree, the production and net margin values remained stable and even exceeded the average values established in the SBHP 2015/21, reaching the maximum values in the years 2015 and 2017. This shows that the activation of the measures and the mobilisation of the resources have enabled the continuance of one of the principal economic engines of the Segura Basin, even during a period of drought with scarce resources.

## 6. Conclusions

As analysed in this study, drought is a natural, cyclical phenomenon and the consequence of a reduction in the rainfall of a region, which puts the capacity to meet the demands at risk and, as a result, is prone to generating impacts on human activities.

The Segura River Basin, located in south-east Spain, is one of the regions which has historically suffered most due to this phenomenon which, together with a permanent scarcity situation, has led to serious consequences for the environment and socioeconomic activities. In turn, in view of climate change predictions, the risk and also the resilience capacity of the exploitation systems are forecasted to increase.

In spite of all of this, significant progress has been made to address this phenomenon, including actions such as the mobilisation of non-conventional resources, the reuse of treated wastewater and the desalination of seawater, together with savings policies and special drought plans which have helped to anticipate the detection through the use of indicators and the activation of measures in accordance with the different scenarios considered.

The relevance of this analysis resides in the revision made of the Special Drought Plan approved in 2007, with important changes in the definition of the methodology to apply and the incorporation of new elements in the Special Drought Plan of 2018. It should be noted that the change in the regulations occurred during the last period of drought recorded in the Segura Basin between the years 2015 and 2019.

After analysing the methodologies of both Special Drought Plans, we have compared the date of the approval and the duration of the Drought Decrees which could have been approved in both situations, reaching the following conclusions:

- Through the SDP 2007, the conditions for activating the Drought Decree were fulfilled when the transfer system indicator entered a situation of alert (0.325) in January 2015. Therefore, the Drought Decree could have been approved four months earlier.
- The period during which the Drought Decree was in force could have been extended until May 2018, when the transfer system indicators shifted to a pre-alert situation (0.368), having a total validity of 40 months.
- Given that the Drought Royal Decree 356/2015 was in force from 8 May to the end of the fourth extension on 30 September 2019, according to the SDP 2007 indicators (in force during the drought period analysed), there was a four-month delay in the approval of the aforementioned decree and a delay in its finalisation of more than one year.
- Through the SDP 2018 (not in force in the approval of the Drought Decree and the four extensions), two periods with the scenarios required for the activation of the Drought Decree would have arisen: the first in January 2016 when the global system indicator showed a situation of alert of scarcity (0.239), coinciding with prolonged drought in the headwaters of the Tajo (0.197); and the second in October 2016 when the global system indicators showed a situation of alert (0.262), coinciding with prolonged drought in the headwaters of the Tajo 0.278).
- With the conditions of the SPD 2018, the Drought Decree in the first case would only have been in force for the month of January 2016, as the drought in the headwaters of the Tajo would have returned to a situation of normality in February 2016. In the second case, the Drought Decree would have been in force from October 2016 to May 2018 (19 months) when the global system reached the situation of pre-alert (0.428).

Consequently, with the new methodology of the SDP 2018, the period during which the Drought Decree was in force would have reduced by half, to only 19 months, which would have led to a delay in activating the measures to mitigate the lack of resources to meet the demands and reduce the impacts both on supplying society and on the principal economic engine: agriculture.

As already mentioned, the value of the area's agricultural sector is fundamental and undisputed not only in the Segura Basin, as it accounts for a high proportion of Spain's agro-food exports, having the highest productivity of the Iberian Peninsula. The figures calculated in the latest planning studies estimate the production value associated with irrigation in the Segura Basin at €3000 M/year and the net margin at €1400 M/year, with more than 115,000 jobs generated.

Therefore, as reflected in Table 17, during the period of validity of the Drought Decree (approved through the methodology of the SDP 2007), the maximum volumes authorised of extraordinary resources to mitigate the negative effects of the drought of 2015–2019 exceeded 630 hm³. However, with the methodology of the SDP 2018, they could have been reduced considerably, leading to an increase in the total deficit of the basin, calculated at 434 hm³/year (Table 8), principally in the agricultural sector which needs more than 80% of the basin's total resources.

Despite this situation of a fragile equilibrium prevailing in the Segura Basin in situations of normality, the drought has not affected agriculture in terms of the production value and net margin. Thanks to the detection systems (indicators) and measures activated (pursuant to the Drought Decree), the mobilised extraordinary resources have not only been able to maintain the economic values prior to the drought but have increased them in some of the years analysed (Tables 18 and 19).

This fact seems to indicate that with the new methodology developed in the SDP 2018, the periods of drought were adjusted with more precision and, as a result, the exact

moment when the extraordinary resources are needed is more evident; although, this may mean that they have to be reduced at certain times.

In future situations of drought, both the mechanisms developed for detection in the SDP 2018 and, principally, the activation of the measures should be tested in order to verify whether they adjust more precisely to the needs that are generated.

Finally, it should be noted that the drought phenomenon constitutes one of the most important challenges in terms of the management of water resources on an international level. Climate change is aggravating and increasing the frequency of these phenomena which has necessitated the undertaking of exhaustive studies in order to anticipate and respond to them. The Special Drought Plans developed in the Segura Basin constitute an example for other basins, both in Spain and internationally, of the advances made in the study of this phenomenon. Furthermore, as shown in this document, they enable the impacts to be mitigated and maintain the supply to the population and the competitiveness of the principal productive systems.

**Author Contributions:** J.A.R.-O. wrote the presented manuscript, compiled and processed private and public data of this study. M.A.S.-G. and M.I.L.-O. designed the study and revised the manuscript, provided private data and references (literature). All authors have read and agreed to the published version of the manuscript.

**Funding:** This research was funded by the Water Chair of the University of Alicante-Alicante Provincial Council (2022) and by the Campus Hábitat5U network of excellence.

**Institutional Review Board Statement:** Not applicable.

**Informed Consent Statement:** Not applicable.

**Data Availability Statement:** Data are contained within the article.

**Conflicts of Interest:** The authors declare no conflict of interest.

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
