# Peer review of "Procedures and Legal Instruments for Drought Declaration in the Segura River Basin (Spain)"

_water, doi:10.3390/w14142171_

Round 1

Reviewer 1 Report

The present article is focused on analyzing the last drought occurring in the Segura River Basin in the period 2015-2019.

The abstract doesn't include any information on the methodology that has been used, achieved results and which is the main conclusion. Thus, this part needs to be revised.

The methodology that has been used is not clearly presented. It is rather very difficult to be followed. 

In addition, it is not clear which is the relevance of this study at European and international level.

Why 9-month SPI is relevant as drought indicator for your study?

The Results, Discussions and Conclusions sections are widely described but not sufficiently linked. The results are not sufficiently/ clearly addressed in the discussions section.

Reference list should be revised since it doesn't entirely follows the MDPI rules.

Author Response

Thank you for reviewing the study and your helpful comments. Modifications have been made in the document in response to the comments received. Below you will find a justification of the changes made and their identification in the final document.

Reviewer 2 Report

Recommendations to authors

- Line 2: I suggest you consider changing the title for the manuscript: “Mechanisms and thresholds of progressive severity phases for predicting and detecting drought situations, in the Segura River Basin (Spain)”.

- Line 21: “.. and to verify how the indicator system developed by the Drought Plans has allowed many actions to be implemented so to minimize the adverse effects on the productive systems.”. Where in the text is this mentioned or analyzed? I think you must re-write the last part of the abstract and make the purpose of current work clear. If the purpose is to define and propose mechanisms for predicting and detecting drought situations, and thresholds of progressive severity phases of droughts (normality, pre-alert, alert, and emergency), then it should be added in Abstract, Introduction and Conclusions.

- Line 95: “The research analyses the drought indicators developed in the first SDP and its subsequent review carried out in 2018”. What are the conclusions of this review? What has current study to offer more compared to this review carried out in 2018?

- Line 233, Figure 3: Considering the average annual rainfall in Figure 3, there are two “peaks” for periods 2016-17 and 2018-19, where the rainfall seems to be high enough, although these periods are marked as “drought” periods. I think you should comment on this in combination with what is mentioned in line 550: “the pre-alert threshold was reached in June 2016”. Also, in line 589, Figure 9: for the same period (June 2016), based on the graph, the Segura Index is above the Prolonged Drought Index value and Tajo Index has also high values. Please also comment on this.

- Line 258: Chapter number has a strikethrough format and must be corrected.

- Line 424: Please make the translations of text in the equation in English.

- Line 682, Table 13: Actions carried out in each year are listed but no comment is given apart from that drought indices got worse and continued to decline considerably. It would be interesting to comment on the actions taken or make comparisons between the years. Do authors believe that these actions were adequate or incomplete? What actions/measures should be taken into consideration to anticipate similar situations in the future?

- Line 868: The study ends somehow abruptly; authors must make clear what new this work offers and what the next steps should be. 

- Lines 868-872: This last paragraph must be re-written to emphasize the importance of the findings of current work; you should mention that the above proposed mechanisms and the thresholds of progressive severity phases for predicting and detecting drought situations can lead to a better management in the future and can contribute to future Special Drought Plans in the Segura River Basin (Spain)”.

Author Response

(The authors gave the same response as above.)

Round 2

Reviewer 1 Report

Indeed, the article has been significantly improved. 

However, the EU and international relevance of your study is still not sufficiently explained.

Please add a couple of phrases extending the explanations that you provide in the letter to the reviewers.

Overall, with a minor revision, the article can be accepted.

Author Response

Thank you for reviewing the study and your helpful comments. Modifications have been made in the document in response to the comments received. Below you will find a justification of the changes made and their identification in the final document.

Point 1: However, the EU and international relevance of your study is still not sufficiently explained. Please add a couple of phrases extending the explanations that you provide in the letter to the reviewers.

Response 1: Thanks for your comments, we have added the following text to the document to try to broaden its international relevance, as well as references to new campaigns at European scale in the field of water scarcity:

Lines 74-84: “The phenomena of drought and scarcity constitute one of the most important challenges in international water policy, even more so with the exacerbation occurring due to climate change. In the review conducted of this article, these aspects have been examined and related to the Segura River Basin and may be extrapolated to other international basins.”

Water scarcity already affects every continent as water use has been growing globally at more than twice the rate of population increase in the last century, and an increasing number of regions are reaching the limit at which water services can be sustainably delivered, especially in arid regions and growing urban areas [25]. Climate change is also expected to amplify the already complex relationship between world development and water demand [26].

Line 95-99: “Recently, new campaigns have been launched in the European Union that focus on the increase in water scarcity, not only in arid and semi-arid places, with potentially devastating consequences on a global scale if nothing is done about the impact enough to reverse the situation and increase the risk of the progress to ensure availability and sustainable management of water and sanitation (Sustainable Development Goals) [27].”

Reviewer 2 Report

Recommendations to authors

 - Line 237: Please consider changing the title of Figure 3; instead of “Drought period 2015-2019” you can use: “Net contributions regulated in the headwater reservoirs” or simply “Net contributions (2012-2020)”. You can also just remove the title of this figure.

- Line 612: I think there is no need to use colors in this figure; you can use only black color for the three lines in the graph. Just select different dash types for the two indices.

- Line 659, Figure 13: This figure is overloaded and therefore difficult to read. If possible, please consider keeping only one index or else change the format of the lines so the graph becomes easier to understand (see the above comment).

Author Response

Thank you for reviewing the study and your helpful comments. Modifications have been made in the document in response to the comments received. Below you will find a justification of the changes made and their identification in the final document.

Point 1: Line 237: Please consider changing the title of Figure 3; instead of “Drought period 2015-2019” you can use: “Net contributions regulated in the headwater reservoirs” or simply “Net contributions (2012-2020)”. You can also just remove the title of this figure.

Response 1: Line 253 (figure 3): Thanks for your comments, I have modified the title of figure 3, the new title is “Net contributions regulated in the headwater reservoirs (2012-2020)”.

Point 2: Line 612: I think there is no need to use colors in this figure; you can use only black color for the three lines in the graph. Just select different dash types for the two indices.

Response 2: Line 627 (figure 9): Following your suggestions I have used a single color and selected different types of dashes.

Point 3: Line 659, Figure 13: This figure is overloaded and therefore difficult to read. If possible, please consider keeping only one index or else change the format of the lines so the graph becomes easier to understand (see the above comment).

Response 3: Line 674 (figure 13): I have followed the suggestions received to make it easier to understand the information in the figure and changed the format of the lines (following the criteria of the previous comment).
